# FEEDBACK FRICTION: LLMs Struggle to Fully Incorporate External Feedback

**Dongwei Jiang**[*†]   **Alvin Zhang**[*†]   **Andrew Wang**   **Nicholas Andrews**   **Daniel Khashabi**

djiang21@jhu.edu    bzhang90@jh.edu

Johns Hopkins University

[*]Equal contribution    [†]Corresponding authors

## Abstract

Recent studies have shown LLMs possess *some* ability to improve their responses when given external feedback. However, it remains unclear how effectively and thoroughly these models can incorporate extrinsic feedback. In an ideal scenario, if LLMs receive near-perfect and complete feedback, we would expect them to *fully* integrate the feedback and reach correct solutions. In this paper, we systematically investigate LLMs' ability to incorporate feedback by designing a controlled experimental environment. For each problem, a solver model attempts a solution, then a feedback generator with access to *near-complete* ground-truth answers produces targeted feedback, after which the solver tries again. We evaluate this pipeline across a diverse range of tasks, including math reasoning, knowledge reasoning, scientific reasoning, and general multi-domain evaluations with state-of-the-art language models including Claude 3.7 with extended thinking. Surprisingly, even under these near-ideal conditions, *solver models consistently show resistance to feedback*, a limitation that we term FEEDBACK FRICTION. To mitigate this limitation, we experiment with sampling-based strategies like progressive temperature increases and explicit rejection of previously attempted incorrect answers, which yield improvements but still fail to help models achieve target performance. We analyze FEEDBACK FRICTION and find that models' confidence on specific questions, measured by semantic entropy, predicts feedback resistance: high-confidence predictions remain resistant to external correction. We hope that highlighting this issue in LLMs will help future research in self-improvement.

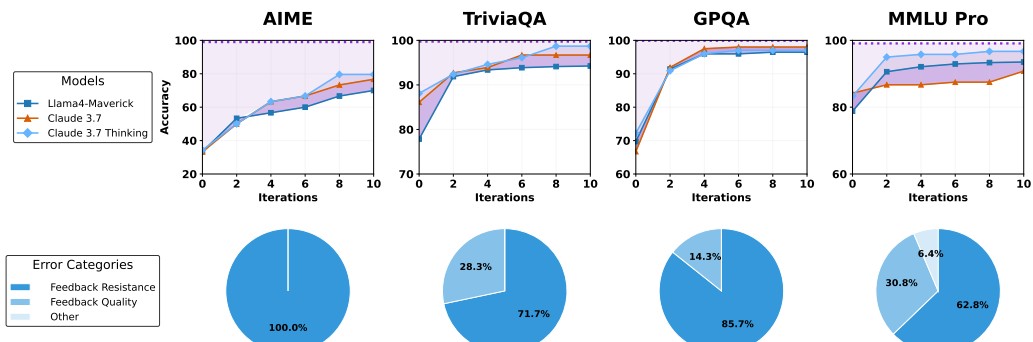

Figure 1: **Top:** Accuracy of various *solver* models when iteratively exposed to feedback from a *feedback* model (GPT-4.1 mini) with access to ground-truth answers. The horizontal dotted line ⋯⋯ represents the target accuracy models could theoretically attain if they successfully incorporated all valid feedback, excluding those with identified issues (details in §4.1). Despite receiving high-quality feedback, **solver models consistently plateau below their target accuracy**. **Bottom:** Breakdown of questions that remained unsolved by the strongest solver model (Claude 3.7 Thinking) after multiple self-correction attempts. **Feedback resistance is responsible for the majority of persistent errors.**

39th Conference on Neural Information Processing Systems (NeurIPS 2025).

# 1   Introduction

The prospect of self-improving Large Language Models (LLMs) has sparked both excitement and debate. Questions persist about LLMs' inherent ability to self-improve without external guidance [1, 2], yet several studies have shown that LLMs can boost their performance when provided with accurate external feedback during test time without any parameter updates or training [3–5]. However, while prior studies establish the *existence* of performance gains from external feedback, *the upper bounds* of such improvement remains largely unexplored. The extent of this improvement would have far-reaching implications for applications such as scientific discovery [6–8] and complex planning [9, 10], where repeated refinement cycles with feedback are critical for success. This raises a critical question: *can models fully incorporate correct feedback to self-improve and reach their maximum potential?*

Answering this question is not straightforward, as self-improvement performance depends on two connected factors: *feedback quality* and *the ability to incorporate feedback*. To decouple these two factors, we create a controlled experimental environment that provides near-optimal conditions for feedback incorporation—one where models receive high-quality, targeted guidance based on complete ground-truth information. Figure 2 demonstrates our setup. The *solver* model attempts to solve problems iteratively, receives feedback from a strong *feedback* model on each incorrect answer, and retries again for up to 10 consecutive iterations. The feedback generator has access to the complete history of all previous attempts and responses, enabling targeted guidance that addresses persistent errors.

While this controlled environment with high-quality feedback provides an ideal testing ground, the effectiveness of feedback incorporation may depend significantly on the *quality* of the feedback. To assess them, we implement three increasingly sophisticated feedback mechanisms to determine how much feedback quality impacts model performance and whether any level of feedback enables models to reach target accuracy. Binary Correctness Feedback ($F_1$) provides simple indication of correctness (e.g., "the answer is wrong"). Self-Generated Reflective Feedback ($F_2$) has the model itself analyze potential errors using correct answers and available solution steps (e.g., "you correctly set up the equation but made an error when computing the derivative..."). Finally, Strong-Model Reflective Feedback ($F_3$) uses a more capable external model (GPT-4.1 mini) to generate feedback. Details of our evaluation framework and feedback generation process can be found in §2.

We conduct a systematic study using all three forms of feedback across strong frontier models including Llama-3.3-70B-Instruct, Llama-4-Scout-17B-16E-Instruct, Llama-4-Maverick-17B-128E-Instruct-FP8, Claude 3.7 Sonnet, and Claude 3.7 Sonnet with Extended Thinking. We evaluate these models across diverse tasks, including AIME 2024, MATH-500, TriviaQA, PopQA, MMLU, MMLU Pro, GPQA, and two synthetic digit multiplication tasks. While higher-quality feedback does improve self-improvement performance, a fundamental limitation persists. As shown in Figure 1, even with our best feedback mechanism—Strong-Model Reflective Feedback ($F_3$)—models consistently fall short of the target accuracy (i.e., the accuracy of an ideal model if it successfully incorporated all the given feedback). We name this weakness FEEDBACK FRICTION (§3.2).

Our analysis in §4 reveals several key insights into FEEDBACK FRICTION. First, we categorize errors that persist after multiple feedback iterations and find that *feedback resistance* (i.e., models failing to incorporate clear and accurate feedback) *is the dominant failure mode across all tasks* (§4.1). Second, we attempt to mitigate it by avoiding repeated wrong answers through sampling strategies. While performance improves across tasks, models still fall below their target accuracy (§4.2). Finally, we investigate why models resist feedback (§4.3). Our investigations rule out several apparent causes, including reasoning complexity, data familiarity, and whether specific questions consistently challenge all models. However, we uncover two key findings: First, using semantic entropy to measure confidence, we find that less confident models show greater relative improvements from feedback, indicating confident models are more resistant to correction. Second, models consistently claim to understand feedback and express willingness to update their beliefs (>95%) yet fail to actually incorporate corrections—revealing a disconnect between stated intentions and actual behavior. In summary, having shown that state-of-the-art LLMs consistently resist external feedback, we identify important patterns in this resistance while ruling out several intuitive explanations.[1]

---

[1]Code for this work is available at: `https://github.com/JHU-CLSP/Feedback-Friction`

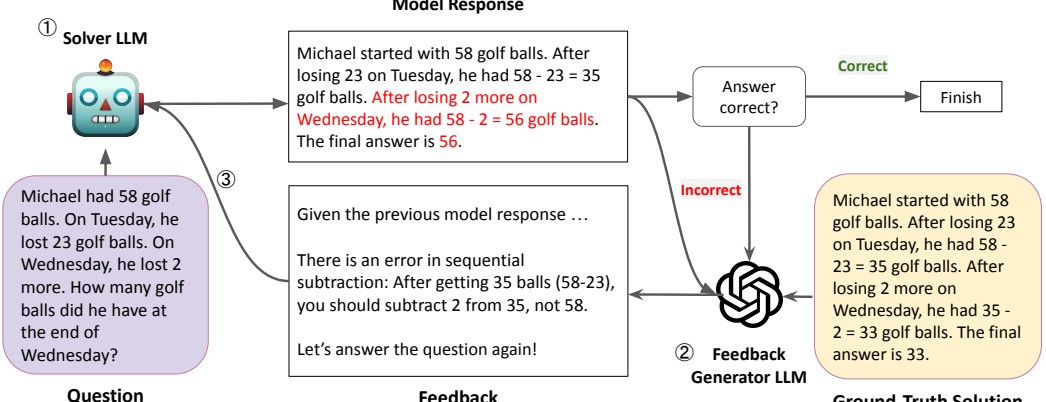

Figure 2: **Iterative self-improvement loop.** The process involves: ① a *solver* model generating an answer, ② a *feedback* model generating feedback given incorrect responses and the ground-truth correct answer, and ③ the *solver* attempting again with this feedback. This cycle repeats for up to 10 iterations or until a correct answer is produced.

## 2 A controlled framework to surface FEEDBACK FRICTION

Our framework employs two key components: an iterative self-improvement loop that allows models multiple opportunities to correct their mistakes (§2.1), and a spectrum of feedback mechanisms with varying levels of detail and guidance (§2.2).

### 2.1 Setup for iterative self-improvement loop

Given a task $T$ with evaluation dataset $D = \{(x_i, y_i)\}_{i=1}^{m}$ and evaluation metric $f$, we establish an iterative improvement protocol with two distinct models: a solver model $M_{\text{solver}}$ and a feedback generator model $M_{\text{feedback}}$.

For each input $x_i$, the solver model produces an initial answer $a_1(x_i)$ using the standard task prompt. We evaluate the correctness of this answer using $f(a_1(x_i), y_i)$, where $y_i$ is the ground truth. If the answer is incorrect, the feedback generator $M_{\text{feedback}}$ creates targeted guidance $g_1$ based on the current answer and ground-truth information.

For iteration $k \geq 1$, we construct the prompt $p_{k+1}(x_i) = \text{concat}(x_i, \text{history}_k)$, where $\text{history}_k$ contains all previous answers and feedback pairs $\{(a_1, g_1), (a_2, g_2), ..., (a_k, g_k)\}$. This process repeats for up to $k = 10$ iterations or until the correct answer is generated. In our empirical experiments, we observed that performance improvements tend to plateau within 10 iterations, suggesting a practical upper limit for the iterative refinement process.

The overall accuracy for the dataset at iteration $k$ is measured as the fraction of all problems solved correctly: $\text{Acc}_k = \frac{1}{m} \sum_{i=1}^{m} \mathbf{1}[f(a_k(x_i), y_i) = 1]$, where $\mathbf{1}[\cdot]$ is the indicator function.

By construction, since we iterate over the incorrect responses only, the accuracy sequence $\{\text{Acc}_1, \text{Acc}_2, ..., \text{Acc}_K\}$ is monotonically non-decreasing, as we retain correct answers across iterations and only modify incorrect ones. This protocol, illustrated in Figure 2, provides a controlled environment for measuring how effectively models incorporate feedback while maintaining consistent evaluation criteria throughout the improvement process.

### 2.2 Designing different feedback mechanisms for iterative self-improvement

We investigate three distinct feedback mechanisms for the self-improvement process, each offering progressively greater guidance and error specificity. All mechanisms are designed to identify errors *without directly revealing the correct answer*, ensuring a fair evaluation of the model's ability to incorporate feedback.

**Binary Correctness Feedback ($F_1$).** The simplest form provides only correctness information:

$$F_1^{\text{binary}}(x_i, y_i) = \text{"The answer is wrong!"} \quad \text{if } f(a(x_i), y_i) = 0$$

where $a(x_i)$ is the solver model's answer and $f$ is the evaluation function.

**Self-Generated Reflective Feedback ($F_2$).** In this approach, the solver model analyzes its own response using available information:

$$F_2^{\text{self}}(x_i, y_i) = M_{\text{solver}}\big(\text{concat}(x_i, a(x_i), y_i, s_i, p_{\text{prompt}})\big)$$

where $p_{\text{prompt}}$ is the instruction: "Please give me feedback on which solution step is wrong and how to get to the correct answer without revealing the answer." Here, $s_i$ represents the ground-truth solution process when available. For datasets without detailed solutions, only the answer $y_i$ is provided.

**Strong-Model Reflective Feedback ($F_3$).** We employ a more capable external model with access to the same information:

$$F_3^{\text{strong}}(x_i, y_i) = M_{\text{strong}}\big(\text{concat}(x_i, a(x_i), y_i, s_i, p_{\text{prompt}})\big)$$

where $M_{\text{strong}}$ represents a more powerful model than $M_{\text{solver}}$ in providing feedback.

Examples illustrating the various feedback types can be found in Appendix A).

# 3 Experimental results

In this section, we present comprehensive experimental results evaluating FEEDBACK FRICTION. We first describe our experimental setup, including tasks, prompts, inference setups, and model configurations (§3.1). We then demonstrate how models consistently plateau below target performance regardless of the feedback mechanisms employed (§3.2).

## 3.1 Experimental setup

**Tasks and metrics.** We employ nine diverse tasks for evaluation, deliberately choosing objective tasks with clear ground-truth answers to ensure reliable evaluation of feedback incorporation.[2] Our tasks include: AIME 2024 [11] and MATH-500 [12] for mathematical problem-solving, TriviaQA [13] and PopQA [14] for knowledge reasoning, MMLU [15] and MMLU Pro [16] for multi-domain evaluation, GPQA [17] for complex scientific reasoning, and two synthetic digit multiplication tasks. The first synthetic task involves 5-digit multiplication (e.g., 78934 × 62851), while the second task applies hexadecimal multiplication rules to decimal numbers (i.e., first mapping 0-9 to themselves and 10-15 to their hexadecimal digits ($10 \rightarrow A$, ..., $15 \rightarrow F$), then performing the arithmetic as if operating in base-16—creating a counterfactual [18] arithmetic setting that challenges models' learned numerical reasoning patterns. For both tasks, ground-truth solutions are generated using deterministic templates that break down the multiplication into smaller multiplication and addition operations. We include these synthetic tasks specifically to remove potential confounding variables from semantic context (more details about these tasks can be found in Appendix D).

Other than the two synthetic tasks, AIME 2024, GPQA, and MATH-500 include complete solutions, while the others provide only final answers. For MMLU and MMLU Pro, the multiple-choice format raises concerns about whether models might solve problems through simple elimination strategies rather than genuine reasoning (e.g., selecting A in the first iteration, B in the second, C in the third, etc). However, our analysis reveals that models exhibit surprising choice persistence even when provided with corrective feedback. Rather than systematically eliminating options or switching between choices, models often remain anchored to their initial selections (typically one or two specific answer choices) across multiple feedback iterations, even when that choice is demonstrably incorrect. We use the same prompts, few-shot demonstrations, answer parsing mechanism, and metrics from `lm-evaluation-harness` and `llama-evaluate` where applicable to maintain established evaluation practices and enable fair comparison with prior work. For MMLU, MMLU Pro, PopQA, and TriviaQA, we sample 10% of the total data points, as running the full dataset through our 10-iteration improvement process would be prohibitively time-consuming. Our preliminary experiments confirmed that this subset yields performance metrics nearly identical to those obtained from the complete dataset.

**Prompt design and experimental controls.** Our experimental framework employs carefully structured prompts to ensure consistent feedback delivery across iterations. For the solver model,

---

[2]Using another LLM to evaluate more subjective tasks like instruction following or translation could lead to issues like reward hacking and unreliable assessments.

we use task-specific system prompts (detailed in Appendix B) and construct iterative prompts that include complete interaction history. We investigate whether prompt structure and organization affect feedback incorporation by comparing our standard single-prompt approach against a multi-turn conversation format that structures the same information across multiple dialogue exchanges. In the conversation format, we reformatted the iterative improvement process to mimic natural dialogue, with alternating turns between solver attempts (as user) and feedback responses (as assistant). Results showed marginal differences compared to single-prompt formatting, suggesting that FEEDBACK FRICTION persists regardless of interaction structure.

Beyond the feedback mechanisms discussed in §2.2, to maintain evaluation integrity while preserving feedback quality, we implement comprehensive answer masking to prevent feedback from directly revealing ground truth solutions. We use "[masked]" as the replacement token for filtered content, applying targeted masking that preserves intermediate steps and reasoning guidance while preventing direct answer revelation. For example, we replace numerical answers with "[masked]" (e.g., "The final answer is [masked]" instead of "The final answer is 42") while preserving solution steps.[3]

**Models and inference.**    As for *solver models,* we employs strong models including LLaMA-3.3 70B Instruct [19], Llama-4-Scout-17B-16E, Llama-4-Maverick-17B-128E-Instruct-FP8 [20], Claude 3.7 Sonnet and Claude 3.7 Sonnet with extended thinking [21]. Claude 3.7 with extended thinking is a variant that employs an extended reasoning before generating final responses, allowing the model to engage in more deliberate problem-solving through explicit step-by-step thinking. All models are instruction-tuned versions of their respective base models, specifically optimized for handling natural language instructions and maintaining consistent output formatting.

For the Llama models, during inference, we use temperature 0 to ensure deterministic outputs and conduct inference using vLLM [22] with each model's corresponding chat template. All inference is performed on a single H100 instance equipped with eight 80GB GPUs. For Claude models, we access them through Anthropic's API. For Claude 3.7, we use temperature 0 to ensure deterministic outputs, while for Claude 3.7 with extended thinking, we use temperature 1 as suggested by Claude. We also explore the effects of varying these temperature settings in our experiments to mitigate FEEDBACK FRICTION (§4.2).

For *feedback models,* we use different models depending on the feedback type. For Strong-Model Reflective Feedback ($F_3$), we utilize GPT-4.1 mini [23] as the feedback generation model. From our internal testing, GPT-4.1 mini's feedback performance is on-par with Claude 3.7. Due to the higher cost of Claude 3.7, we use GPT-4.1 mini for generating the strongest feedback. We also considered o4-mini [24] as the feedback generator model due to its reportedly superior reasoning capabilities, but our experiments showed it delivered comparable feedback quality while incurring substantially higher computational costs, leading us to proceed exclusively with GPT-4.1 mini.

### 3.2   Main findings

**FEEDBACK FRICTION persists across model scales and tasks.**    Figure 3 shows results using our strongest feedback mechanism (Strong-Model Reflective Feedback ($F_3$)) across all datasets and all models. Our results reveal a striking pattern: Despite receiving high-quality feedback, all solver models consistently plateau below their target accuracy, which is the accuracy of an ideal model if it successfully incorporates all the given feedback (see §4.1 for calculation details). Claude 3.7 Thinking achieves the highest initial accuracy on several tasks (AIME, TriviaQA, GPQA, and MATH-500), while Claude 3.7 shows competitive performance across most benchmarks. However, both Claude variants exhibit the same fundamental plateauing behavior as the Llama models. The performance typically improves rapidly through the first 2-4 iterations before significantly slowing down. This FEEDBACK FRICTION is particularly pronounced on complex reasoning tasks like AIME and GPQA, where even the best-performing models remain 15-25% and 3-8% below their respective theoretical ceilings despite 10 correction opportunities. The synthetic tasks reveal particularly interesting patterns across model families. In the standard 5-Digit Multiplication task, both Claude models reach near-perfect accuracy after significant initial improvement, outperforming the Llama models. However, the Hexadecimal-5-Digit-Multiplication task reveals extreme difficulty with feedback incorporation

---

[3]We verified that models do not attempt to predict the "[masked]" token during inference

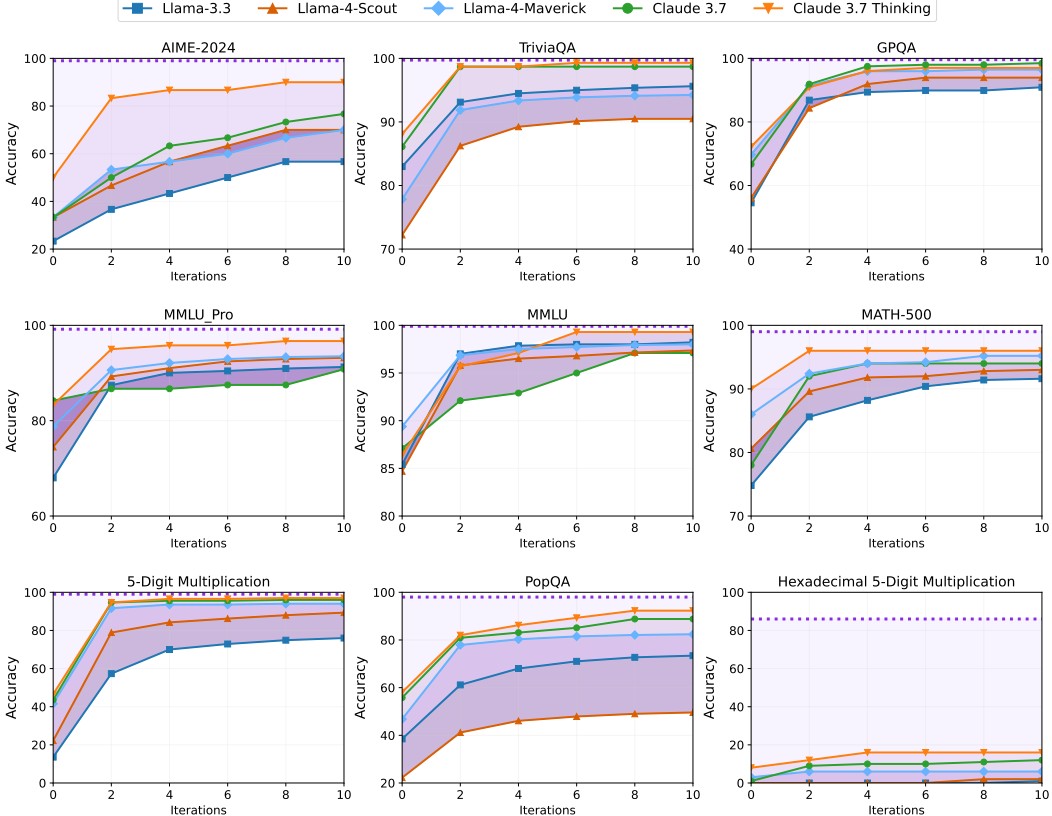

Figure 3: The performance of frontier models we tested with Strong-Model Reflective Feedback ($F_3$) across nine different tasks. Models are given multiple attempts with feedback that incorporates both the final answer and complete solution (when available). The dotted line ···· represents the target accuracy that models could theoretically achieve if they fully incorporated all feedback (details in §4.1). **Results demonstrate that despite strong feedback, models consistently plateau below their target accuracy across all tasks.**

across all models—no model exceeds 20% accuracy even after 10 iterations, highlighting severe limitations in feedback integration for counterfactual arithmetic systems.[4]

**While feedback quality determines the achievable performance ceiling, the extent of its impact is task dependent.** Figure 4 illustrates how different feedback mechanisms affect performance across models and tasks. Due to the cost of running extensive experiments with Claude models, we focus the feedback quality analysis on the Llama model families. All tasks show clear benefits from increasingly sophisticated feedback.

The impact of high-quality feedback is most pronounced on complex reasoning tasks. For AIME, MMLU Pro, and GPQA, Strong-Model Reflective Feedback ($F_3$) outperforms binary feedback by significant margins across all models. Llama-4-Maverick shows the strongest overall performance, achieving 73.3% accuracy on AIME and 96.5% on GPQA with Strong-Model Reflective Feedback ($F_3$)—improvements of +26.7% and +10.6% over binary feedback, respectively. Llama-4-Scout demonstrates the largest relative gains, particularly on AIME (+33.3%) and GPQA (+13.1%), compared to Llama-3.3 which shows more modest improvements (+26.7% on AIME, +6.6% on GPQA). Nevertheless, despite these substantial improvements, all models still plateau significantly below their theoretical performance ceiling.

---

[4]While it is true that models' imperfect understanding of hexadecimal arithmetic leads to imperfect feedback quality, this is reflected in the lower theoretical ceiling shown in the figure. Even accounting for this limitation, models still fall short of what they could achieve if they fully incorporated the available feedback.

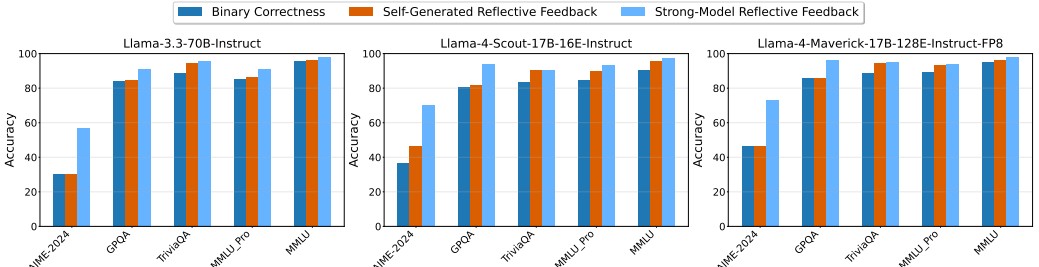

Figure 4: Performance comparison across benchmark datasets using different feedback mechanisms with Llama-3.3, Llama-4-Scout and Llama-4-Maverick. **Model performance progressively improves as feedback quality increases from Binary Correctness Feedback ($F_1$) to Strong-Model Reflective Feedback ($F_3$).**

# 4 Analysis of FEEDBACK FRICTION

We conduct a deeper analysis to better understand FEEDBACK FRICTION. We first categorize different cases where models fail to correct their mistakes despite multiple rounds of feedback (§4.1), then examine the extent to which we can alleviate this problem with sampling strategies (§4.2), and finally, we present several hypotheses and the experiments to understand FEEDBACK FRICTION (§4.3).

## 4.1 Feedback integration failures dominate persistent self-improvement errors

**Error category development.** We manually examine cases where LLMs fail to improve despite receiving high-quality feedback and identifies three main categories of error: (1) Most critically for our analysis, **feedback resistance failures** represent cases where models fail to accurately incorporate feedback despite multiple iterations. (2) **Feedback quality issues** encompass cases where the provided feedback is incorrect, ambiguous, or fails to address the key problematic steps in the solution. This can still occur because the generated feedback might miss crucial conceptual errors or introduce new inaccuracies, even though ground-truth is provided to the feedback generator. (3) We maintain an **"Other"** category for cases that don't clearly fit into either of the above categories. From our initial examination, this includes cases where the problem itself contains ambiguities or the solution is conceptually correct but fails due to style or formalization issues (e.g., providing the correct information but not in the expected format required by the evaluation metric).

**Automated error categorization and validation.** To systematically categorize errors at scale, we used OpenAI o4-mini as an automated annotator to classify complete self-improvement trajectories from Claude 3.7 and Claude 3.7 Thinking according to these predefined categories. To validate this automated approach, we conducted rigorous manual verification by randomly sampling 50 errors from each task and having two human annotators independently label them according to our defined categories.

Table 1: Distribution of error categories (%) of unsolved problems after 10 iterations of self-improvement classified by o4-mini. FR: Feedback Resistance, FQ: Feedback Quality, OTH: other issues.

| Dataset | Solver Model | FR | FQ | OTH |
|---------|--------------|------|------|-----|
| MMLU Pro | Claude 3.7 | 64.6 | 28.0 | 7.4 |
| | Claude 3.7 Thinking | 62.8 | 30.8 | 6.4 |
| GPQA | Claude 3.7 | 100.0 | 0.0 | 0.0 |
| | Claude 3.7 Thinking | 85.7 | 14.3 | 0.0 |
| TriviaQA | Claude 3.7 | 72.4 | 25.0 | 2.6 |
| | Claude 3.7 Thinking | 71.7 | 28.3 | 0.0 |
| AIME 2024 | Claude 3.7 | 100.0 | 0.0 | 0.0 |
| | Claude 3.7 Thinking | 100.0 | 0.0 | 0.0 |

Our verification showed 96% agreement between human annotators and o4-mini's classifications, significantly higher than the 78% agreement rate achieved with GPT-4.1 mini on the same samples. This confirms o4-mini's reliability for this analysis task.

**Feedback resistance dominates error patterns.** Table 1 presents the distribution of error categories across different tasks, as classified by o4-mini. Feedback resistance is consistently the dominant category across all tasks, accounting for 62.8-100% of errors. This finding suggests that the core challenge in achieving perfect performance lies not in the quality of feedback or problem complexity, but in fundamental limitations of how models process and incorporate corrective feedback. Detailed examples of each error category are provided in Appendix C.

## 4.2 Mitigating FEEDBACK FRICTION with sampling strategies

Given the persistent plateau in performance we observed across models and tasks, a natural question arises: *can we mitigate* FEEDBACK FRICTION *through existing strategies?* We explore sampling techniques as a potential solution to help models overcome their apparent resistance to feedback.

**Progressive temperature increases show modest effectiveness.** We first explore temperature-based sampling strategies where the sampling temperature increases with the iterations: 0.0 for iteration 0, 0.15 for iteration 1, 0.3 for iteration 2, and so forth. While other schedules (e.g., exponential increase, fixed higher temperature) could be explored, we chose this linear progression as a simple baseline that gradually introduces diversity while preserving early deterministic behavior. We hypothesize that progressive temperature increases would help models generate more diverse outputs, allowing them to escape from local optima in their output distributions and become more receptive to feedback.

Due to the cost of running extensive experiments with Claude models, we focus the feedback quality analysis on Llama-4-Scout and Llama-4-Maverick. As shown in Figure 5, this approach alone produced minimal improvements compared to the baseline in Figure 3. Analysis of logs revealed that while increased temperature successfully diversified model outputs, the additional exploration often failed to converge on correct answers due to the vast search space of possible responses.

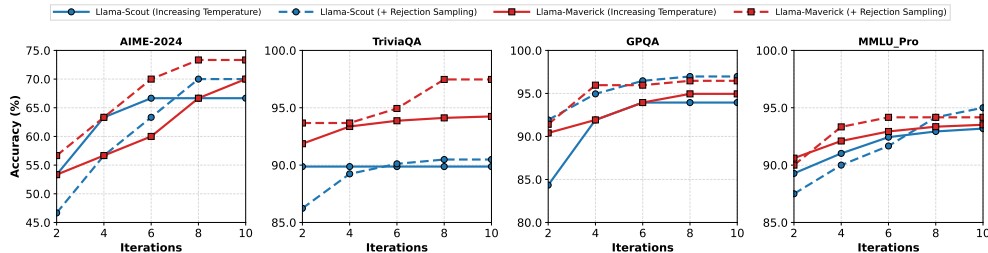

Figure 5: Results of using progressively increasing temperature and rejection sampling with Llama-4-Scout and Llama-4-Maverick. **Rejection sampling can provide additional improvements over temperature-based sampling alone across both multiple-choice and non-multiple-choice tasks.**

**Combining temperature increases with rejection sampling yields better results.** To enhance performance further, we implement a more targeted approach that combines increased temperature with rejection sampling. This method explicitly forces the model to explore new solution paths while avoiding previous attempts. Specifically, we instruct the model to generate 25 answers, and remove final answers that occurred in previous iterations (which by construction is incorrect, since we only continue iterating on problems that remain unsolved). If no answer remains after this filtering process, we randomly select one from those 25 answers. Otherwise, we randomly select one of the remaining novel answers as the final prediction.

As shown in Figure 5, the combined strategy yields substantive performance gains across both multiple-choice datasets and non-multiple-choice datasets compared to the baseline that only increases the temperature. While the magnitude of these improvements varied, the consistent pattern suggests that forcing models to explore new solution paths by rejecting previously used answers is beneficial. Despite these gains, we observed that all datasets still fall short of the target accuracy, indicating that sampling strategies alone cannot fully resolve model resistance to feedback.

## 4.3 Understanding FEEDBACK FRICTION

Our sampling strategies (§4.2), though promising, did not eliminate FEEDBACK FRICTION entirely. Developing more effective interventions requires a deeper understanding of the fundamental causes. In this section, we investigate several hypotheses for why models resist incorporating feedback despite multiple correction opportunities. Throughout our analyses, we provide error bars. They represent the standard error of a binomial proportion, calculated as $\sqrt{\text{acc} \cdot (1 - \text{acc})/n}$, where $n$ is the number of samples in each evaluated group.

**Model confidence vs. FEEDBACK FRICTION.** Could excessive model confidence explain resistance to feedback in FEEDBACK FRICTION? To test this, we measure confidence using *semantic entropy* [25], a method that captures uncertainty at the meaning level rather than surface form

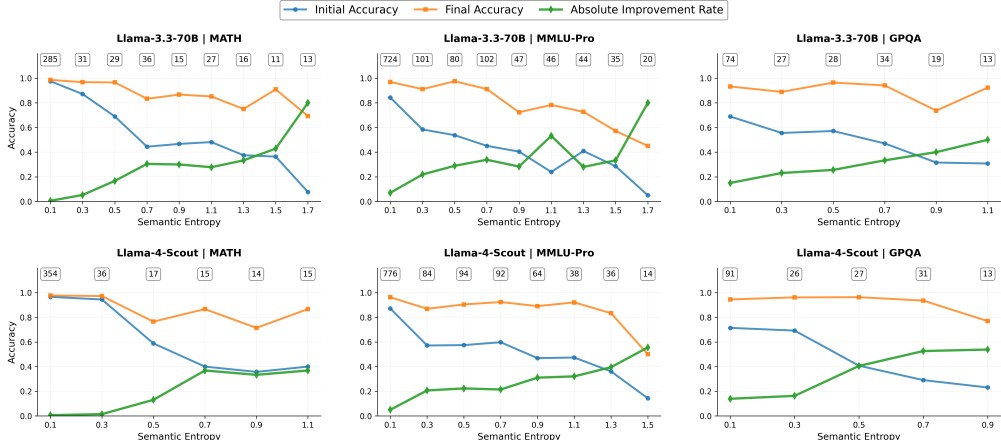

Figure 6: Relationship between semantic entropy and feedback incorporation across three benchmark tasks (MATH, MMLU-Pro, GPQA) for Llama-3.3-70B and Llama-4-Scout models. Semantic entropy (x-axis) measures model uncertainty, with lower values indicating higher confidence. Three metrics are shown: initial accuracy before any feedback (blue), final accuracy after iterative feedback (orange), and absolute improvement rate calculated as $(Final - Initial)/(1 - Initial)$ (green). Numbers in boxes indicate sample sizes per bucket. The absolute improvement rate generally increases with semantic entropy across all tasks and models, indicating that models with lower initial confidence (higher entropy) show greater relative improvements from feedback.

variations. Unlike token-level probability measures, semantic entropy accounts for the fact that models can express the same meaning in multiple ways, providing a more robust measure of true uncertainty.

To compute semantic entropy for each model response, we first generate multiple outputs $\{s_1, s_2, \ldots, s_n\}$ by sampling the model $n = 50$ times with temperature 0.7, where each $s_i$ represents a complete answer given the feedback. We then cluster these outputs using final answers, which we find to have similar results as semantic equivalence grouping using bidirectional entailment. For each semantic cluster $C_i$, we estimate its probability as $\hat{P}(C_i|x) = \sum_{s \in C_i} P(s|x) / \sum_{j=1}^{K} \sum_{s \in C_j} P(s|x)$, where $x$ represents the input (question, previous answer, and feedback), and compute semantic entropy as $H = -\sum_i \hat{P}(C_i|x) \log \hat{P}(C_i|x)$. Lower entropy indicates the model consistently generates semantically equivalent answers (high confidence), while higher entropy suggests diverse semantic meanings (low confidence).

We conduct this analysis across 5 digits multiplication, GPQA, MATH, MMLU-Pro, and TriviaQA using semantic entropy with bucket size 0.2. Figure 6 shows some of the results (full results in Appendix J) and reveals a consistent pattern: absolute improvement rate (the green trends) increases with semantic entropy, rising from near-zero at low entropy (high confidence) to 0.4-0.8 at higher entropy levels (low confidence). These results suggest that models with lower confidence (higher semantic entropy) have both more room for improvement and greater receptiveness to feedback, while highly confident models show minimal improvement despite receiving the same quality feedback.

**Model self-perception versus actual behavior.** To probe deeper into FEEDBACK FRICTION mechanisms, we directly asked solver models about their understanding of feedback and willingness to incorporate it. After receiving feedback, we prompt models with: (1) "Do you understand this feedback?" and (2) "Will you update your belief or understanding about this problem?"

Surprisingly, across all models and tasks, solver models consistently claimed to understand the feedback ($> 95\%$ "yes" responses) and expressed willingness to update their beliefs ($> 96\%$ "yes" responses). However, in subsequent iterations, these same models failed to actually incorporate the feedback. This disconnect between stated intention and actual behavior reveals a fundamental issue: models exhibit self-assessment failure where they believe they are incorporating feedback while demonstrably failing to do so (the prompts used for probing and example conversations can be found in Appendix I). This disconnect between stated intention and actual behavior reveals a fundamental implementation failure: models express both understanding and willingness to change but fail to execute these changes in practice.

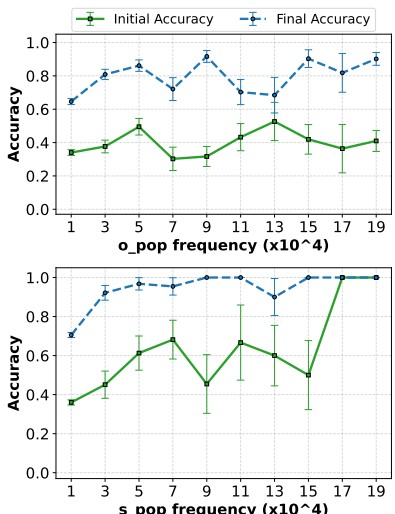

Figure 7: Accuracy of Llama3.3 on PopQA with respect to s_pop and o_pop

**Data familiarity and FEEDBACK FRICTION** Prior work [14, 26] suggests that language models perform better with familiar entities and topics encountered frequently during training. Are these models more resistant to feedback about familiar entities? We investigated whether this familiarity bias contributes to FEEDBACK FRICTION using PopQA's popularity metrics as proxies for entity familiarity. This dataset includes monthly Wikipedia page views for subject entities $s_{prop}$ and object entities $o_{prop}$. We then analyzed how accuracy changes during iterative self-improvement correlate with the popularity of the subject ($s\_prop$) and object ($o\_prop$) entities.

We use Llama-3.3 for this experiment since it was released closer to the publication date of PopQA. These popularity metrics provide a particularly relevant measure of potential training data frequency for this model. As shown in Figure 7, we found no consistent pattern between entity popularity and accuracy. We provide further supporting evidence in Appendix F with additional statistical testing and alternative popularity metrics.

**Further analyses.** We conducted additional analyses to investigate whether problem complexity or inherent question characteristics correlate with feedback integration failure. Both hypotheses yielded negative results. Complete details are provided in Appendix G and Appendix H.

# 5 Related Work

**Self-Improvement with LLMs.** Self-improvement in AI has roots in early work using GANs for self-generated feedback [27, 28]. LLMs have expanded these capabilities across code generation [29], reasoning [30 **?** , 31], instruction following [32–34], and other domains [35–37]. Approaches vary between training time improvements through self-correction [38] or data generation [39, 40], and inference time improvements via feedback [3, 30] or other models [41]. While studies question intrinsic self-improvement without external feedback [1, 2], consensus exists that LLMs can self-improve with ground-truth feedback [5, 42]. In this work, we probe these limits and investigate barriers to full feedback integration.

**The elasticity–plasticity tension.** LLMs face a trade-off between retaining prior knowledge and integrating new information. In continual learning, they exhibit catastrophic forgetting when updates conflict with existing beliefs [43–45]. In model editing, extensive edits distort broader knowledge networks [46, 47]. Alignment techniques like RLHF instill desired behaviors but models remain tied to pre-trained distributions or revert via adversarial prompts [48, 49]. In agentic settings, LLMs fail to adapt when encountering external failures [50]. Conversely, they show excessive plasticity through sycophantic behavior in conversations [51]. We revisit this dilemma through feedback integration during self-improvement and unveil controlling factors.

**Feedback for LLMs.** Feedback mechanisms include intrinsic self-evaluation [52–55] and extrinsic feedback via tools [56–58], information sources [59, 60], or ground-truth answers [3, 61]. While correct feedback is key [5, 62], LLMs struggle to incorporate it when contradicting prior knowledge [63], handling refutations [64], assessing uncertainty [65], or facing misleading feedback [66, 67]. Different from previous work, in this paper, we focus on high-quality feedback, investigating why LLMs fail to fully incorporate it.

# 6 Conclusion

Our study reveals a fundamental limitation in LLMs' ability to incorporate external feedback. Despite receiving high-quality feedback over multiple iterations, models consistently plateau below their theoretical performance ceiling across diverse reasoning tasks. Despite extensive analysis, the precise mechanisms underlying feedback resistance remain elusive, which we leave to future work. Understanding and addressing these limitations remain essential for developing more adaptable AI systems capable of genuine and sustained self-improvement.

## Limitations

**Better understanding of FEEDBACK FRICTION** While our study identifies key insights into FEEDBACK FRICTION—including the correlation between semantic entropy and feedback resistance and the disconnect between models' stated understanding and actual behavior, we lack a definitive mechanistic explanation for why models resist incorporating feedback. A better understanding of FEEDBACK FRICTION likely involves complex interactions between feedback understanding, instruction following, and belief updating. Future work would benefit from more sophisticated mechanistic interpretability techniques, such as causal intervention methods and circuit analysis [68, 69], to understand the specific computational pathways through which feedback resistance emerges and persists across model architectures.

**Limited mitigation strategies** Despite extensive experimentation with sampling strategies, our attempts to fully mitigate feedback friction yields only modest improvements. Recent work has explored techniques to enhance diversity in LLM outputs through reinforcement learning [70] and training-free prompting strategies [71], which may help models generate more varied responses to feedback. Another promising approach is to perform supervised fine-tuning or reinforcement learning that could (1) enhance the solver model's receptiveness to feedback incorporation, and (2) improve the feedback generator's ability to provide more effective guidance [72]. However, the computational constraints of our experimental setup, particularly the large model sizes we tested (including 70B+ parameter models like Claude 3.7 and Llama-4-Maverick) and the associated computational costs, prevented us from conducting fine-tuning experiments that might meaningfully address feedback resistance. We leave that exploartion to future work.

## Acknowledgment

This work is supported by ONR grant (N00014-241-2089). The GPUs were provided by the DSAI and ARCH clusters. We sincerely thank Yuqing Yang, Zhengping Jiang, and the broader JHU community for discussions and feedback.

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

# A   Examples of different feedback types

We provide a complete worked example illustrating the feedback types differences in our study in Figure 8. The problem (AIME 2024-I-10) involves finding a segment length in a circle with tangent lines. The solver incorrectly assumes the triangle is right-angled and applies an unjustified formula.

F2 identifies two specific formula errors: using cos A instead of sin A, and using angle (A+C) instead of B in the Law of Cosines. F3 provides six numbered issues with extensive explanations and a step-by-step correction roadmap.

---

**Original Question:**
Let ABC be a triangle inscribed in circle ω. Let the tangents to ω at B and C intersect at point D, and let AD intersect ω at P. If AB=5, BC=9, and AC=10, AP can be written as m/n, where m and n are relatively prime integers. Find m + n.

**Solver Model's Attempt:**
Step 1: We need to find the length of AP, where AD intersects ω at P. We should utilize properties of inscribed angles, tangents to circles, and similar triangles.
Step 2: Recall that the tangent to a circle is perpendicular to the radius drawn to the point of tangency. By using the properties of tangents, we can relate the lengths of segments BD, CD, and AD.
Step 3: Apply the power of a point theorem. For point D, we have DB · DC = DA · DP.
Step 4: To find DB and DC, use the fact that DB and DC are tangents to the circle and BC is a chord. Let x = BD = CD and use the given side lengths AB = 5, BC = 9, and AC = 10.
Step 5: Given AB = 5, BC = 9, and AC = 10, we notice that 5² + 9² = 10², which means △ABC is a right-angled triangle with ∠BCA = 90°.
Step 6: With △ABC being a right triangle, we can find the radius R of circle ω using R = abc/(4Δ), where Δ is the area. The area Δ = (1/2) · 5 · 9 = 45/2, and R = (5 · 9 · 10)/(4 · 45/2) = 5.
Step 7: Since the triangle is right-angled at C, we focus on using BC, AC, and AB to find AP through areas and similar triangles.
Step 8: By Power of a Point from D, DB · DC = DA · DP. Also, note that DA = DP + AP.
Step 9: We can use the fact that △ABD and △ACD are similar to △PBD and △PCD respectively, or directly apply Power of a Point and similarity to find AP.
Step 10: For a right-angled triangle inscribed in a circle, the mean proportional theorem can be applied. Using AB · AC = AD · AP or similar relations, we compute AP.
Step 11: Direct computation of AP using AB = 5, BC = 9, AC = 10. From the relation, AP = (AB · AC)/(AB + AC) = (5 · 10)/(5 + 10) = 50/15 = 10/3.
Step 12: To get the sum m + n where AP = m/n, we have m = 10 and n = 3, so m + n = 10 + 3 = 13.

The final answer is: 13

**Binary Correctness Feedback (F1):**
The answer is wrong!

**Self-Generated Reflective Feedback (F2) Using the Solver Model Itself:**
Here are the critical errors in your reasoning:

1 **Expression for CD**
You set CD = (BC∕2) / cos A. In fact, in isosceles ΔBDC (with base BC and base-angles A), BC = 2·CD·sin A, so CD = BC∕(2 sin A), not BC∕(2 cos A).
2 **Law of Cosines on ΔACD**
You wrote AD² = AC² + CD² − 2·AC·CD·cos(A + C). But ∠ACD is not A + C. By the alternate-segment theorem (or inscribed-angle chase) one finds ∠ACD = ∠ABC = B.
Hence the correct relation is AD² = AC² + CD² − 2·AC·CD·cos B.

These two mis-steps propagate through your arithmetic for AD, DP and then AP. Once you replace BC/(2 cos A) with BC/(2 sin A) to find CD, and use ∠ACD = B in the LoC for AD, the subsequent power-of-a-point calculation DP = CD²/AD and then AP = |DP − AD| will fall into line and yield the true rational value of AP.

**Strong-Model Reflective Feedback (F3) Using GPT-4.1:**
Thank you for sharing the solution and your reasoning! Here's some feedback identifying the issues in the previous approach and guidance on the steps to correct:
1 **Step 5 and 6 (Right triangle and radius):** The claim that the triangle is right angled because 5² + 9² = 10² is actually incorrect in this problem. Since 5² + 9² = 25 + 81 = 106 which is not equal to 10² = 100, the triangle is **not** right angled. This means the calculations that follow assuming a right angle, including the radius computation based on right triangle properties, cannot be used.
2. **Step 7 (Using incorrect triangle configuration):** The reasoning in Step 7 mistakenly treats the triangle as right angled and misapplies the Pythagorean theorem to find lengths related to points B,D,C. So, lengths BD, DC, etc., cannot be simplified as done there.
3. **Step 11 (Formula for AP):** The step that writes AP = (AB · AC)/(AB + AC) is not a known or justified direct formula and comes without proof or reference. Without the triangle being right angled or a direct similarity/proportionality argument, such a formula does not hold here.
4. **Ignoring the angle relationships from tangent lines:** The most crucial fact is that the intersection point D of the tangents at B and C satisfies ∠BDC = ∠BAC (the tangents form an angle equal to the arc they subtend). This fact is a cornerstone to finding exact expressions for lengths AD, CD, and ultimately AP.
5. **Missing Law of Cosines computations to find AD and CD:** To correctly solve the problem, use the Law of Cosines in △ACD and △BCD, incorporating the angle relationships from the tangents and arcs to express AD and CD in terms of known sides and cosines of angles like ∠A and ∠B.
6. **Power of a Point at D:** Applying the power of a point theorem at D is key, but DP × AD = DB × DC. Since DB = DC (both are tangent lengths from point D to the circle), this relation helps you solve for DP. Then AP = AD - DP.

These corrections will lead to the proper final expression for AP.

---

Figure 8: Examples of different feedback types.

# B  Prompts and evaluation details for problem solving and feedback generation

## B.1  Prompts used for the solver model

We developed different prompting strategies for the solver model to incorporate feedback and generate new solutions across iterations. The system prompts in Figure 9 were used for the solver model across different tasks:

```
Math-500 and AIME 2024:
You are a smart assistant that solves math problems. Please think step by step to solve the problem. If you think you're ready
to output the answer, you can wrap your answer with \\boxed{}. Please follow this format

TriviaQA:
You are a smart assistant that solves trivia questions. If you think you're ready to output the answer, you can just output an
answer.

MMLU and MMLU Pro:
The following are multiple-choice questions about {category}. Let's think step by step. Please explain your reasoning clearly as
you work toward the answer. When you're ready, conclude your answer with the phrase: \"The answer is (X)\" where X is the
correct letter choice. Make sure to always include parentheses around the letter.

GPQA:
The following are multiple-choice questions. Let's think step by step. Please explain your reasoning clearly as you work toward
the answer. When you're ready, please finish your answer with \"The answer is (X)\" where X is the correct letter choice. Make
sure to always include parentheses around the letter.

PopQA:
You are a smart assistant that answers fact-based questions. If you think you're ready to output the answer, you can just output
an answer.

5-Digit Multiplication:
You are a smart assistant in solving multiplication questions. Please think step by step. If you think you're ready to output
the answer, you can wrap your answer with \\boxed{}. Please follow this format.

Hexadecimal 5-Digit Multiplication:
You are a smart assistant in solving hexadecimal multiplication questions. Please think step by step. If you think you're ready
to output the answer, you can wrap your answer with \\boxed{}. Please follow this format.
```

Figure 9: System prompts used for the solver model across all tasks.

At the initial generation round, we directly use the question as the prompt for the solver model. For multiple-choice questions, we format the question by concatenating the question and answer choices with their corresponding labels (i.e., A to D for MMLU and GPQA, A to J for MMLU Pro). In subsequent rounds, we provide the solver model with its complete previous history and the corresponding feedback from the feedback generator model. We clearly label each iteration so the solver model can track all its previous attempts. The general template for the iterative prompt structure is provided in Figure 10.

```
Question:
[Original Question]

Prompt for Generation after the Initial Round:
You are given the full history of your previous attempts and the feedback provided for each attempt.
History:

Attempt at (iteration 1) and the corresponding feedback:

Answer: [answer1] Feedback: [Feedback 1 Depends on Feedback Strategy].

Attempt at (iteration 2) and the corresponding feedback:

Answer: [answer2] Feedback: [Feedback 2 Depends on Feedback Strategy].

Question: [Original Question]. Please answer the question again.

* [answer1] and [answer2] corresponds to the model's first two failed attempts. [Original Question] refers to the initial question the
model attempts to answer.
```

Figure 10: Prompt used for iterative self-improvement

## B.2  Evaluation details for problem solving

We employ few-shot prompting across all tasks to provide consistent context for the solver model. For TriviaQA and PopQA, we randomly sample 5 questions without replacement as few-shot examples. For MMLU and MMLU Pro, we similarly sample 5 questions from the corresponding question category to ensure domain-relevant examples.

For PopQA, we employ an LLM-as-a-judge approach [73] to assess answer correctness. This is necessary because PopQA provides limited answer aliases (extensive alternative phrasings for exact string matching) compared to TriviaQA. Without this approach, models would be penalized for minor formatting differences rather than genuine comprehension errors, leading to an underestimation of their true problem-solving capabilities. For other tasks, we follow the same evaluation metrics provided by lm-eval-harness.

## B.3 Prompts used for feedback generation

We implement three distinct feedback generation strategies as described in §2.2. For Binary Correctness Feedback ($F_1$), we provide minimal information: "Your answer was incorrect. Please answer the question again."

For Self-Generated Reflective Feedback ($F_2$) and Strong-Model Reflective Feedback ($F_3$), we employ identical prompt templates that differ only in the model used for generation. The feedback generator receives the complete interaction history, including all previous solver attempts and corresponding feedback. When available, we provide the feedback model with detailed solution explanations that justify the correct answer; for datasets lacking such explanations, we provide only the ground truth answer. This approach ensures the feedback model has sufficient context to generate targeted, informative guidance while maintaining consistency across feedback types, and its template is shown in Figure 11.

```
Prompt for Generating the Feedback:
There was a mistake in answering the following question:

[Original Question]

You are provided with the full history of previous attempts made by a separate model, along with corresponding feedback.
History:
Attempt at (iteration 1) and the corresponding feedback:

Answer: [answer1] Feedback: [Feedback 1 Depends on Feedback Strategy].

Attempt at (iteration 2) and the corresponding feedback:

Answer: [answer2] Feedback: [Feedback 2 Depends on Feedback Strategy].

Most Recent Answer: [answer3]

The correct final answer is: [Ground Truth Answer]
The correct reasoning process that leads to this answer is: [Solution with Process for this Question]

Please provide feedback identifying which step(s) were incorrect or how to get to the correct answer WITHOUT revealing
the correct final answer or the content of the correct option.

* [answer1] [answer2] [answer3] refer to all previous attempts the model had. [answer3] is the most recent attempt.
[Ground Truth Answer] is the answer for this question. [Solution with Process for this Question] is the detailed solution
provided for the question in the dataset.
```

Figure 11: Prompt used for generating the feedback.

## B.4 Answer masking in feedback

To ensure fair evaluation, we implement comprehensive answer masking to prevent feedback from directly revealing ground truth solutions while preserving feedback quality. Our approach allows feedback to contain detailed solution steps and guidance but strictly prohibits explicit disclosure of final answers. We use "[masked]" as the replacement token for filtered content.

**Multiple-choice questions.** We mask all possible representations of the correct choice letter. For example, if the correct answer is A, we filter variants including (A), \boxed{A}, **A**, etc.

**Open-ended questions.** For TriviaQA, we filter all terms matching the words in "aliases" and "normalized aliases" answer fields. For PopQA, we mask entries from the "possible answers" answer field. For mathematical tasks (5-digit multiplication and MATH-500), we mask standalone numerical answers and those in \boxed{} notation. Hexadecimal multiplication follows similar patterns. For multiplication tasks, we additionally mask intermediate partial products to prevent reduction to simple addition problems (detailed in Appendix D).

## B.5 Error Categorization Prompt

The prompt template used for categorizing persistent model errors after 9 iterations is shown in Figure 12.

```
System Prompt:
You are an error categorizer specialized in analyzing why Language Learning Models (LLMs) "
fail to self-improve when solving problems. When provided with an LLM's prediction trajectory and the feedback it receives,
you will categorize the errors into one of six categories:
1. Problem is Impossible to Solve
   - The problem itself is fundamentally flawed
   - External tools are required (e.g., calculator for complex calculations, search engine for obscure facts)
2. Problem is Too Complicated
   - The problem exceeds the model's knowledge scope
   - Example: A level 5 math problem beyond the model's training
3. Feedback is Wrong
   - The feedback generator model provides incorrect guidance
   - The feedback fails to identify actual mistakes in the model's response
   - The feedback is too vague or generic to be helpful
4. Model is Not Following Feedback
   - The model fails to incorporate feedback properly
5. Style or Formalization Issue
   - Logical correctness but formatting or notation problems
6. Unknown
   - Cannot categorize the failure into above categories
End your response with exactly: 'The error is: [category]' where category is one of:\n- impossible to solve\n- too
complicated\n- feedback is wrong\n- model is not following the feedback\n- style or formalization issue\n- unknown

User Prompt:
The question and the interaction between the agent model and the feedback model is: {question}
The process answer is: {process_answer}
Please categorize the error. End your response with:\n'The error is: [category]

* {question} is the first 5 iterations of the answer-feedback pairs; {process_answer} is the best ground truth solution we can
provide from the dataset to the model.
```

Figure 12: Error Categorization

## C   Error categorization examples from iterative self-improvement

This section presents representative examples of persistent errors that prevent models from achieving correct solutions despite multiple feedback iterations. We illustrate the main error categories identified in our analysis: feedback resistance (where models fail to incorporate valid corrections, see Figure 13) and feedback quality issues (where the provided guidance is incorrect or misleading, see Figure 14).

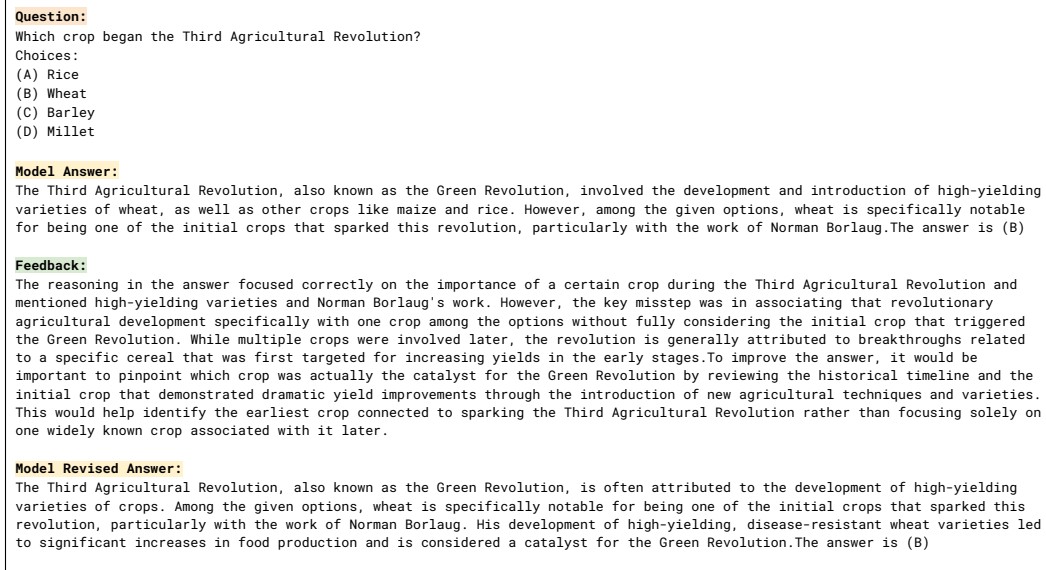

Figure 13: Llama-4-Scout resisting feedback from GPT-4.1 mini in MMLU

```
Question:
A homeowner entered into a contract with an insurance company to purchase a fire insurance policy for her new home. The policy
provided that the insurance company promised to pay up to $500,000 if the said house was destroyed by fire or fire-related
explosion, while the homeowner promised to pay a quarterly premium of $400. The homeowner failed to make the last two quarterly
payments before her house burned to the ground. The insurance company refused to pay the homeowner for the loss because of her
failure to make the last premium payments. In an action by the homeowner against the insurance company to recover for the loss of
her house, the homeowner's best theory of recovery is that
Choices:
(A) although she failed to make the last two premium payments, there was a bargained-for exchange.
(B) the insurance company's duty to pay was not expressly conditioned on the homeowner's duty to make the payments.
(C) the insurance company was under an independent duty to pay for the loss.
(D) the homeowner did not receive notice of cancellation.

Model Answer:
The answer is (D)

Feedback:
The primary error in the previous reasoning was focusing on the homeowner's lack of notice of cancellation rather than examining
the nature of the contractual obligations and the conditional duties within the insurance policy.\n\nIn analyzing this scenario,
it's important to understand that insurance contracts often contain conditions precedent\u2014the insurer's duty to pay is
typically contingent upon the insured's fulfillment of certain obligations, such as timely payment of premiums.
The mistake was not sufficiently weighing how failure to pay premiums affects the insurer's duty to perform under the contract.
The key is to assess whether the insurance company's obligation is dependent upon the homeowner's premium payment, and if that
obligation exists independently or is conditioned on such payment.
Therefore, the analysis should have involved a closer look at the contractual terms and the interdependence of duties, rather
than focusing chiefly on issues related to cancellation notices.

O4-mini's judgement:
The feedback repeatedly mischaracterized the contract\u2019s structure by insisting the insurer\u2019s duty was a condition
precedent on timely premium payments, thereby directing the model away from the correct theory that the insurer owed an
independent duty to pay. In other words, the guidance itself was incorrect and prevented the model from reaching the right
answer. The error is: feedback is wrong
```

Figure 14: Wrong feedback provided by GPT-4.1 mini judged by o4-mini

# D   Synthetic digit multiplication task details

## D.1   5-digit multiplication

We construct a controlled arithmetic dataset consisting of 450 5-digit multiplication problems following the template: ''Calculate the following question:  19365 × 12534.''

**Feedback generation**    We employ a deterministic, human-designed template based on the distributive property to generate ground truth solutions. This template systematically decomposes each multiplication into partial products, providing a clear step-by-step solution pathway. Our template-based approach serves two key purposes: (1) demonstrating structured problem decomposition strategies for complex arithmetic, and (2) ensuring feedback correctness and interoperability. In Figure 15, we illustrate an example template solution, which serves as the reference for feedback generation. The feedback model compares solver outputs against this structured breakdown to identify specific computational errors.

**Answer masking strategy**    To maintain task difficulty, we also mask intermediate partial answers before providing feedback to the solver model. The final feedback combines the masked template solution with model-generated guidance tailored to the specific errors observed.

```
Question:
Calculate the following question: 32183 * 40672.

Manually Generated Feedback:
The original question is:
Calculate the following question: 19365 * 12534.
Step1: After breaking down the numbers: 19365 = 19000 + 365, 12534 = 12000 + 534
Stwp2: Then we apply the distributive property:
19365 x 12534 = (19000 + 365) x (12000 + 534)
Compute partial products:
Step3: 19000 x 12000 = 228000000
Step4: 19000 x 534 = 10146000
Step5: 365 x 12000 = 4380000
Step6: 365 x 534 = 194910
Step7: Sum all partial products we have the answer: 242579910
Total: 228000000 + 10146000 + 4380000 + 194910 = 242579910
```

Figure 15: Templates for 5-digit multiplication solution.

## D.2   Hexadecimal 5-Digit Multiplication

We also extend our synthetic arithmetic evaluation to hexadecimal multiplication, creating problems that challenge models' ability to work with non-standard number systems. The question tem-

```
Question:
Calculate the following question, where each number is represented in base 16: 69837 * 17635.

Manually Generated Feedback:
Multiplying 69837 by 17635 in base 16:
Step 1: Multiply 69837 by '5' \u2192 Partial product: 20F913, Shifted: 20F913, Intermediate Sum: 20F913
Step 2: Multiply 69837 by '3' \u2192 Partial product: 13C8A5, Shifted: 13C8A50, Intermediate Sum: 15D8363
Step 3: Multiply 69837 by '6' \u2192 Partial product: 27914A, Shifted: 27914A00, Intermediate Sum: 28EECD63
Step 4: Multiply 69837 by '7' \u2192 Partial product: 2E2981, Shifted: 2E2981000, Intermediate Sum: 30B86DD63
Step 5: Multiply 69837 by '1' \u2192 Partial product: 69837, Shifted: 698370000, Intermediate Sum: 9A3BDDD63
Final Product (hex) = 9A3BDDD63
```

Figure 16: Hexadecimal 5-Digit Multiplication process solution.

plate follows the format: ``Calculate the following question, where each number is represented in base 16:  69837 $\times$ 17635.'' All answers are expected in base-16 format.

**Template-based feedback.**    Similar to decimal multiplication, we generate feedback using deterministic step-by-step templates. The multiplication process involves sequentially multiplying the first operand by each digit of the second operand (interpreting digits in base-16), then summing the resulting partial products with appropriate positional shifts. Figure 16 demonstrates an example template solution. We validate solution correctness by verifying that partial product summation matches results from standard base-16 calculators.

**Masking strategy for hexadecimal multiplication.** Given the increased computational complexity of hexadecimal arithmetic, we employ a more permissive masking approach. We mask only the final summation step while preserving intermediate partial products. This design balances providing sufficient guidance with maintaining the core challenge of hexadecimal computation.

## E    Analysis of model confidence and FEEDBACK FRICTION

Figure 17 presents confidence-accuracy relationships across four datasets: GPQA, MMLU, MMLU Pro, and TriviaQA. Each plot displays three metrics: initial accuracy at iteration 0, final accuracy after iterative feedback, and the improvement delta between them.

We define a model's *initial confidence* in its generated answer as the exponential of the average log-probability per token in the answer sequence:

$$\text{Initial Confidence} = \exp\left( \frac{1}{T} \sum_{t=1}^{T} \log p(a_t \mid a_{<t}, q) \right)$$

where:

- $T$ is the number of tokens in the generated answer (with EOS excluded),
- $a_t$ is the $t$-th token in the answer,
- $a_{<t}$ denotes the prefix of the answer up to (but not including) position $t$,
- $q$ is the input question,
- $p(a_t \mid a_{<t}, q)$ is the model's probability of generating token $a_t$ given the previous tokens and the input.

This corresponds to the geometric mean of the per-token probabilities assigned by the model, and serves as a quantitative measure of the model's confidence in its complete answer.

We find that **initial confidence strongly predicts initial accuracy** across all datasets. Higher confidence bins consistently correspond to higher initial performance (Figures 17a–17d), confirming that the model's confidence is a good predictor of its initial accuracy.

However, **confidence poorly predicts improvement potential**. The relationship between initial confidence and accuracy gains varies substantially:

- GPQA shows peak improvements at moderate confidence levels, with diminishing returns at higher confidence

- MMLU and MMLU Pro exhibit relatively flat improvement patterns with the less confident questions getting more improvements. Nevertheless, this may caused by the low initial accuracy.

- TriviaQA demonstrates erratic improvement fluctuations regardless of initial confidence

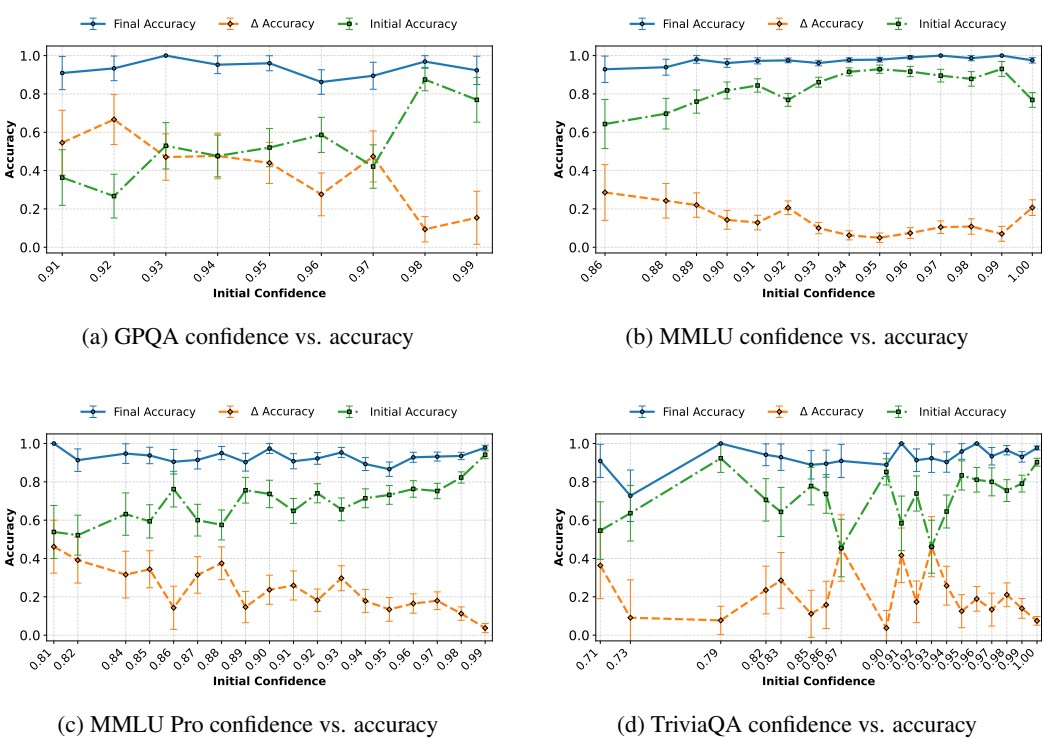

(a) GPQA confidence vs. accuracy

(b) MMLU confidence vs. accuracy

(c) MMLU Pro confidence vs. accuracy

(d) TriviaQA confidence vs. accuracy

Figure 17: Confidence vs. accuracy across different datasets using GPT-4.1 mini as feedback model and Llama-4-Scout as the solver.

# F Analysis of data familiarity and RIGID THINKING

After analyzing data familiarity using answer frequency in the PopQA dataset, we found no clear correlation between model performance and frequency of answer words in the training data. However, surface-level frequency may not fully capture a model's true familiarity with content, as it fails to account for context quality, semantic relationships, and other factors affecting knowledge acquisition during pre-training.

To better capture actual familiarity, we examine a more direct behavioral signal called in-domain performance: the model's accuracy in answering questions, measured using 100 generations per question with Llama-3.3. This behavioral familiarity metric reflects the cumulative effect of all factors contributing to the model's internalized knowledge.

Figure 18 illustrates the in-domain performance of Llama-3.3 across four benchmarks—GPQA, TriviaQA, 5-digit multiplication, and MMLU Pro. We bucket questions based on the model's initial accuracy and report both initial and final accuracy after iterative feedback. While the model shows improvement across all buckets, questions with higher behavioral familiarity (higher initial accuracy) consistently achieve higher final accuracy as well. This suggests that behavioral familiarity is a more informative predictor of both current performance and improvement potential than answer frequency alone. Nevertheless, we still cannot obverse any consistent patterns across all these datasets in the initial vs. final accuracy.

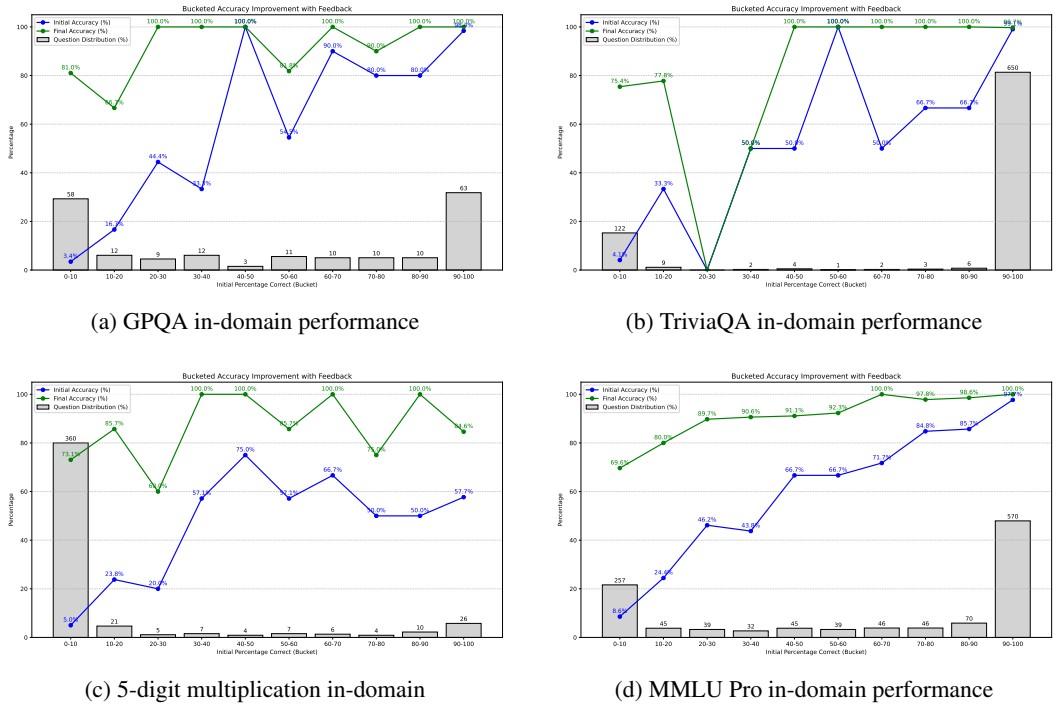

(a) GPQA in-domain performance

(b) TriviaQA in-domain performance

(c) 5-digit multiplication in-domain

(d) MMLU Pro in-domain performance

Figure 18: In-domain accuracy of Llama-3.3 across four benchmark tasks.

# G   Analysis of reasoning complexity and FEEDBACK FRICTION

To investigate whether the model's improvement over iterations correlates with question difficulty or the reasoning complexity, we compare the performance of Llama-4-Scout on two synthesized multiplication tasks: 5-digit and 6-digit problems. Unlike prior datasets, which lack clear separability in difficulty levels, these tasks were manually constructed with 450 questions each to ensure a well-defined difference in complexity.

The initial accuracy of Llama-4-Scout is 2.2% on 5-digit multiplication and 0.889% on 6-digit multiplication. Interestingly, while the 6-digit task is objectively more difficult, we observe greater improvement across iterations compared to the 5-digit task. One possible explanation is that more difficult tasks offer more room for feedback-driven correction because solver model has less initial knowledge about how to solve them. However, we also observe cases where simpler questions yield higher final accuracy, suggesting that the relationship between task complexity and feedback effectiveness is non-monotonic and influenced by additional factors.

# H   Analysis of model type and FEEDBACK FRICTION

To better understand the overlap in failure cases among Llama-3.3, Llama-4-Scout, and Llama-4-Maverick, we compared their incorrect predictions across several benchmark datasets. Specifically, we report the number of shared mistakes between each pair of models, the number of questions all three models got wrong, the total number of unique mistakes (union), and the **Overlap Ratio**, defined as the proportion of all-three common errors to the total number of distinct errors.

The *Overlap Ratio* offers a normalized measure of agreement in model failures. As shown in Table 2, **AIME** shows the highest overlap (35.7%), suggesting a subset of examples that all three models consistently struggle with. Conversely, datasets such as **GPQA** and **5-digit Multiplication** exhibit minimal overlap (6.9% and 0.7%, respectively), indicating that the models tend to fail on different questions.

These findings suggest that model failures are often idiosyncratic rather than being concentrated around a universally difficult subset of examples. The relatively low overlap across datasets highlights the challenge of achieving robust self-correction: errors are not easily attributable to a common set of pitfalls, but rather reflect distinct weaknesses in each model's reasoning and generalization.

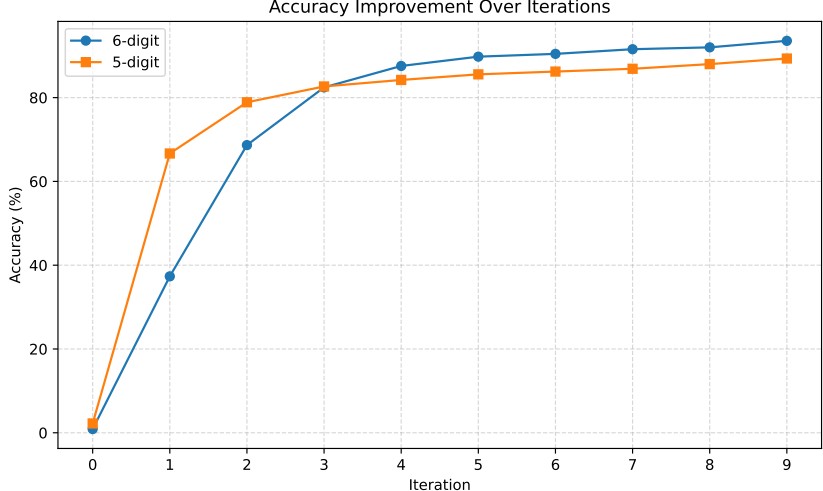

Figure 19: Comparison of the Improvement for 5-digit and 6-digit multiplication with GPT-4.1 mini as feedback model

| Dataset | L3.3–Scout | L3.3–Maverick | Scout–Maverick | All-Three | Union | Overlap Ratio |
|---------|-----------|---------------|----------------|-----------|-------|---------------|
| AIME | 8 | 9 | 5 | 5 | 14 | 0.357 |
| TriviaQA | 22 | 16 | 28 | 14 | 105 | 0.133 |
| MATH-500 | 18 | 14 | 11 | 9 | 64 | 0.141 |
| MMLU | 15 | 12 | 19 | 11 | 55 | 0.200 |
| MMLU Pro | 49 | 40 | 43 | 30 | 163 | 0.184 |
| GPQA | 3 | 5 | 2 | 2 | 29 | 0.069 |
| 5-digit Mult. | 21 | 3 | 1 | 1 | 141 | 0.007 |

Table 2: Pairwise and three-way common failure cases among Llama-3.3, Llama-4-Scout, and Llama-4-Maverick across datasets. Overlap Ratio is computed as the number of questions all three models failed on divided by the union of all distinct failures.

# I  Details of testing model's self-perception of FEEDBACK FRICTION

Figure 20 shows the prompts we used to probe whether the model understand the given feedback and whether it's willing to change its belief.

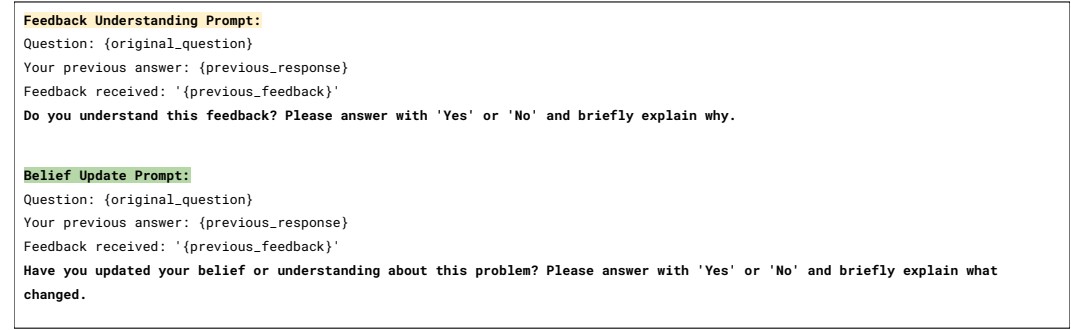

Figure 20: An example of Llama-4-Scout explicitly acknowledging changing its belief while still making the same mistake in the next attempt.

Figure 21 shows an example where Llama-4-Scout explicitly acknowledges that "the actual boundary is defined by the perpendicular bisectors of the segments connecting O to the vertices, which can result in an oblique line" and claims its approach has been updated. Yet in its second attempt, it still describes the region as forming "a square with side length 1/2 min(AB, AD)" and arrives at the same incorrect answer of 1/4. This demonstrates the model can articulate understanding of the feedback while its actual problem-solving behavior remains unchanged. The correct answer is 1/2.

```
Original Question:
Rectangle ABCD has center O and AB/AD=k. A point is randomly chosen from the interior of rectangle ABCD. What is the probability
that it is closer to O than to any of the four vertices?

Model's First Attempt:
The region closer to O than to any vertex is a smaller rectangle with side lengths 1/2 AB and 1/2 AD. This smaller rectangle has
an area of 1/2 AB × 1/2 AD = 1/4 AB × AD. The area of the entire rectangle ABCD is AB × AD. Thus, the probability is 1/4 AB × AD /
AB × AD = 1/4. The final answer is: 1/4

Feedback Provided:
The original solution assumes that the region closer to the center O than to any of the vertices forms a smaller rectangle aligned
with the original rectangle, with side lengths exactly half of the original. However, the locus of points closer to O than to a
particular vertex is not a straight boundary parallel to the sides; rather, it is defined by the perpendicular bisector of the
segment connecting O and that vertex, which is generally an oblique line. The assumption that the smaller region is exactly
one-quarter of the whole area because it is a rectangle of half the original sides is not correct.

Model's Statement on Belief Update:
Yes. My understanding of the problem has changed based on the feedback provided. Initially, I incorrectly assumed the region
closer to the center O than to any vertex formed a smaller rectangle with sides exactly half the lengths of the original
rectangle. However, the feedback highlighted that this assumption is incorrect and that the actual boundary is defined by the
perpendicular bisectors of the segments connecting O to the vertices, which can result in an oblique line. This realization,
combined with the importance of considering symmetry, transformations, and a more nuanced geometric partitioning of the rectangle,
has updated my approach to solving the problem.

Model's Second Attempt:
By drawing these perpendicular bisectors, we can see that the region closer to O than to any vertex in this smaller rectangle
forms a square with side length 1/2 min(AB, AD). The area of this smaller rectangle is 1/4 AB × AD, and the area of the region
closer to O is 1/2 min(AB, AD) × 1/2 min(AB, AD). However, considering the setup and focusing on the probability aspect, we should
approach it by understanding that the division of the rectangle into regions closer to O or the vertices is based on the
perpendicular bisectors, which effectively creates a smaller square (when considering the closest region to O) within each of the
four smaller rectangles. Given the symmetry and the nature of the division, the probability simplifies to the ratio of the area
where points are closer to O than to any vertex, over the total area of the rectangle. This simplification leads to recognizing
that the probability does not depend on the aspect ratio k but rather on the geometric properties of the rectangle and its
division. The final answer is: 1/4
```

Figure 21: An example of Llama-4-Scout explicitly acknowledging changing its belief while still making the same mistake in the next attempt.

## J Full results of semantic entropy

Figure 22 presents the complete self-correction analysis across all five benchmark tasks (5 digits multiplication, GPQA, MATH, MMLU-Pro, and TriviaQA) for both Llama-3.3-70B and Llama-4-Scout. Each subplot shows initial accuracy (blue), final accuracy (orange), and absolute improvement rate (green) as functions of semantic entropy with bucket size 0.2. Regions with fewer than 10 samples are marked with red shading and dashed boundaries to indicate unreliable data, with sample counts displayed in boxes above each plot.

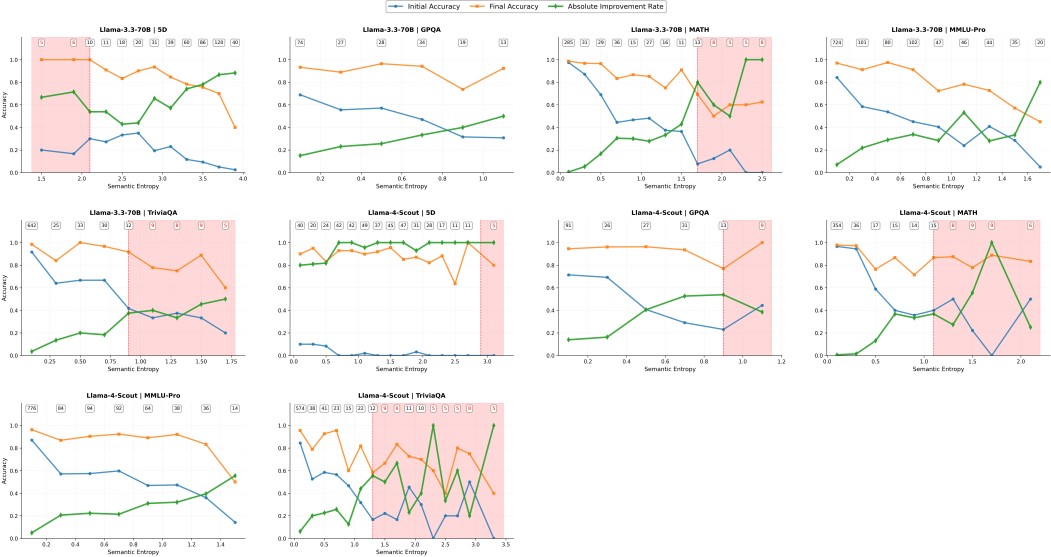

Figure 22: Full results of semantic entropy over five benchmark tasks and two models.

