# OpenReview forum: "FEEDBACK FRICTION: LLMs Struggle to Fully Incorporate External Feedback"
_NeurIPS.cc/2025/Conference — NeurIPS 2025 poster_

### Official Review · Reviewer_xvok · 2025-06-22

**Clarity:** 3
**Significance:** 3
**Originality:** 3
**Rating:** 4
**Confidence:** 5

**Summary:**

This paper examines the tendency of LLMs to not incorporate external feedback. The paper defines the *Rigid Thinking* effect - showing the models' inability to respond to adequate feedback. The results show the quantitative frequency of *Rigid Thinking* across different tasks for 3 feedback strategies and 3 models. It shows that all models reach saturation significantly below the theoretical limit. Ablation studies also claim that the cause of Rigid thinking does not lie with overconfidence and data familiarity. Finally, this work further investigates promising potential mitigations for the effect of *Rigid Thinking*, showing promise for combatting the effect at test time.

**Questions:**

1. Large Reasoning Models, such as DeepSeek R1, Qwen3, o4-mini, etc. have been shown to not only be better at reasoning tasks, but also more resistant to external feedback. It would be interesting if the authors could replicate their experiments, with, if necessary, following the pointers I mentioned in the **Strenghts and Weaknesses** section. Given the computational difficulties, and the rather time-consuming nature of human-based evaluations, some results on a smaller model
such as *DeepSeek R1 Distill Qwen3 8B*, *Qwen3 8B*, or similar models with the same properties, would be helpful for their arguments. I aknowledge that not all of these models were not released at the time of the submission deadline but alternatives, such as the DeepSeek R1 Distill 14B were present.

2. In sampling the LLM generations, the authors describe using a deterministic approach (with temperature 0). Why is that necessary for quantifying the phenomenon, and why are different temperature schedules not considered in the main evaluation? Ideally, some experimental evidence would be appreciated.

3. How would using conversation-based feedback (repeated user-assistant iterations), instead of a large prompt containing all the iterations, change the results?

4. Can the authors address the comments about their confidence comparison I made in **Strengths and Weaknesses**?

5. Why did the authors not consider scaling using automated tools? This would allow for much better scaling across multiple models. In most human-based studies, the experts' judgments are not always 100% accurate, so the 96% accuracy rate of o4-mini seems quite high.

If the authors are able to address these issues, I would be happy to raise my score.

**Ethical Concerns:**

["NO or VERY MINOR ethics concerns only"]

**Final Justification:**

I have increased my score from an initial 2 to a 4 following the strong rebuttal presented by the authors. In particular, they provided a much more thorough evaluation on a proprietary reasoning model, one of the best in terms of robustness - which revealed that RIGID THINKING persists even in near-SOTA models. This makes further investigation of the phenomenon highly relevant.

The authors also addressed the majority of my remaining concerns. They offered a deeper analysis of how RIGID THINKING relates to model confidence, and clarified the questions I had about the experimental setup. The potential for using this work as a benchmark makes it even more compelling.

The only reason I did not raise my score higher is that evaluating a broader range of models would have been a valuable addition  something that could be easily addressed in the next revision. Moreover, uncovering evidence of the potential root causes of RIGID THINKING and identifying training interventions to mitigate it would elevate this work even further.

The fact that the authors addressed so many issues within the two-week rebuttal period is deeply impressive, and I believe it warrants close attention from both the remaining reviewers and the AC. I am happy to advocate for the paper’s acceptance, as the authors have addressed nearly all the concerns raised by myself and other reviewers.

**Limitations:**

The authors have provided some discussion on the limitations of their work. They acknowledge minor limitations, such as not being able to completely remove the Rigid Thinking phenomenon, as well as not being able to explain why models can behave stubbornly.

However, they do not thoroughly discuss the limitations about the scale and replicability of their results for other models and settings. I discuss most of these weaknesses in the **Strengths and Weaknesses** section. In summary:
- There is no clear reason why no competing open-source models (Gemma/Phi), or reasoning models (o4-mini, Qwen3, R1) were not investigated. Instruction-following behaviour might differ vastly across different model types, or differences in how these models were trained.
- The authors should discuss the difficulty of evaluating this dataset, and potential for automated scalability in more detail.

**Paper Formatting Concerns:**

I have no formatting concerns, given that the authors update their Appendix as they did in the supplementary material.

**Quality:**

3

**Strengths And Weaknesses:**

## Strengths
The paper is well explained, with clear experimental setup. Quantifying receptiveness to feedback is an important avenue for research, and proving the majority of the cause is the models' response, rather than the quality of the feedback. The range of the tasks is sensible, including a range of easier, and more challenging ones. Further, the additionally defined tasks are interesting, with the differences in behaviour between hex and decimal computation showing a key limitation in LLMs, not just in the Rigid Thinking sense.

The observation and quantification of the phenomenon is important, as it highlights the need for future research in this direction.

Showing that this effect can be mitigated is also a key contribution. It would be nice to see if train-time techniques can be used to do this, as the proposed test-time solutions can be computationally expensive

With the paper's findings, I hope that better train- or test-time mitigations for "Rigid Thinking" will be developed as a result.
## Weaknesses

The paper is mostly clear and presented well, some decisions for the experimental setup were not thoroughly explained. While the LLMs being evaluated are quite recent, their scope is rather limited. The paper only looks into the most recent LLaMa models, where potentially similar training methods and/or data might have induced similar behaviour. Furthermore, the authors acknowledged that they didn't evaluate on the Qwen3 235B-A22B reasoning model, as it got stuck in reasoning loops. However, in their experimental setting, the authors described using a temperature of 0. This decision is not explained well enough, and is the primary cause for the reasoning loops observed. In particular, basically all open-source reasoning model providers especially warn against using a low temperature. Showing the observations in this work hold across different model providers/reasoning models, would significantly strengthen the paper.

Furthermore, the human-led nature of the study into the factors causing Rigid Thinking can be considerably difficult to scale. The authors show that o4-mini has a near-human performance with a >96\% agreement rate, yet they do not explain why a human-based evaluation was necessary.

Unfortunately, the paper does not present a clear argument about why the effect exists, as well for eliminating the overconfidence factor. First, the measurement use to quantify confidence is not sufficiently justified. It is further temperature-dependent, given the choice of the sequence of generated tokens is as well. Finally, given the final accuracy is near-saturation, the statistical variability around 100% seems much more significant than the confidence factor. Evaluating on a harder task (i.e. AIME) where accuracy is not almost saturated by the final iteration, might give a stronger signal to support the hypothesis.

A nitpick I had with the feedback preprocessing step is the fact that masking the boxed answer is often insufficient. Especially in more reasoning-heavy tasks, intermediate or near-final steps might leak a significant of information.

Overall, the paper contains some good ideas, but the experimental design leaves key questions and issues not throughly explored.

---

> ### Author Rebuttal · Authors · 2025-07-31
>
> We thank the reviewer for recognizing that our paper is well explained with a clear experimental setup, that quantifying receptiveness to feedback is important research, and that our observation and mitigation of this phenomenon represents a key contribution. We have tested additional reasoning models that confirm RIGID THINKING persists, clarified our model-specific temperature settings, and conducted multi-turn interaction experiments. Below we provide detailed responses to your specific concerns:
>
> ---
>
> > The scope of models tested is limited to LLaMA family only, no reasoning model is tested
>
> **TL;DR: We tested Claude 3.7 reasoning models and found the same RIGID THINKING behavior persists.**
>
> Although we did not evaluate reasoning models in our NeurIPS submission, we have since tested larger models like Claude 3.7 Sonnet (both with and without Extended Thinking). Table 1 shows the initial accuracy, final accuracy after 10 feedback iterations, and theoretical target accuracy for each model. Despite Claude models achieving significantly higher performance than Llama models, on several datasets, they still plateau well below their target accuracy on several datasets, demonstrating that RIGID THINKING persists across model scales. We'd definitely welcome suggestions for additional models to test.
>
> | Dataset                    | Solver Model                           | Initial (%) | Final (%) | Target (%) | Δ (Target - Final) |
> |----------------------------|----------------------------------------|------------:|----------:|-----------:|-------------------:|
> | Hexadecimal 5-Digit Mult   | Claude 3.7 Sonnet                      |        1.0 |      12.0 |       86.0 |               74.0 |
> | Hexadecimal 5-Digit Mult   | Claude 3.7 Sonnet (Extended Thinking)  |        8.0 |      16.0 |       86.0 |               70.0 |
> | AIME 2024                  | Claude 3.7 Sonnet                      |       33.3 |      76.7 |       99.0 |               22.3 |
> | AIME 2024                  | Claude 3.7 Sonnet (Extended Thinking)  |       50.0 |      90.0 |       99.0 |                9.0 |
> | PopQA                      | Claude 3.7 Sonnet                      |       55.8 |      88.8 |       98.0 |                9.2 |
> | PopQA                      | Claude 3.7 Sonnet (Extended Thinking)  |       58.1 |      92.3 |       98.0 |                5.7 |
> | MMLU Pro                   | Claude 3.7 Sonnet                      |       84.2 |      90.8 |       99.2 |                8.4 |
> | MMLU Pro                   | Claude 3.7 Sonnet (Extended Thinking)  |       83.3 |      96.7 |       99.2 |                2.5 |
> | MATH-500                   | Claude 3.7 Sonnet                      |       78.0 |      94.0 |       99.0 |                5.0 |
> | MATH-500                   | Claude 3.7 Sonnet (Extended Thinking)  |       90.0 |      96.0 |       99.0 |                3.0 |
> | 5-Digit Multiplication     | Claude 3.7 Sonnet                      |       43.6 |      96.0 |       99.0 |                3.0 |
> | 5-Digit Multiplication     | Claude 3.7 Sonnet (Extended Thinking)  |       46.6 |      97.0 |       99.0 |                2.0 |
> | GPQA                       | Claude 3.7 Sonnet                      |       66.7 |      98.5 |       99.6 |                1.1 |
> | GPQA                       | Claude 3.7 Sonnet (Extended Thinking)  |       72.2 |      97.0 |       99.6 |                2.6 |
> | MMLU                       | Claude 3.7 Sonnet                      |       87.1 |      97.1 |       99.9 |                2.8 |
> | MMLU                       | Claude 3.7 Sonnet (Extended Thinking)  |       86.4 |      99.3 |       99.9 |                0.6 |
> | TriviaQA                   | Claude 3.7 Sonnet                      |       86.1 |      98.7 |       99.7 |                1.0 |
> | TriviaQA                   | Claude 3.7 Sonnet (Extended Thinking)  |       88.1 |      99.3 |       99.7 |                0.4 |
>
> **Table 1:** Performance comparison of Claude 3.7 models showing initial accuracy, final accuracy after 10 feedback iterations, and theoretical target accuracy across different datasets.
>
> ---
>
> > Temperature setting (temperature of 0 is chosen throughout the analysis, even for Qwen models)
>
> **TL;DR: In the current submission, we used appropriate temperatures for each model, not uniformly temperature=0.**
>
> We want to clarify our temperature choices, which were not uniformly set to 0 as the reviewer suggests:
>
> We tested Qwen3-235B-A22B with temperature=0.7 (following the suggested temperature from their documentation), but found that when reasoning mode is enabled, it still enters repetitive generation loops frequently, and it's not very good at following instructions (the community observed a similar phenomenon), which led to our final decision not to use it.
>
> In addition, in our run with Claude 3.7 with Extended Thinking, we use temperature=1 as suggested by Anthropic.
>
> We choose temperature=0 for most of our testing to ensure deterministic results crucial for controlled experimentation. However, we did conduct experiments with higher temperatures in later sections (§4.2), where we explore progressive temperature increases and rejection sampling as potential mitigation strategies.
>
> ---
>
> > Human-led study into factors causing Rigid Thinking would be difficult to scale
>
> Yes, as you pointed out, that is a major roadblock if we were to only rely on human evaluation to under Rigid Thinking! However, we did this time-consuming human evaluation for the exact opposite purpose — to verify that o4-mini's judgment of why LLMs didn't follow instructions is correct, so that we **don't** need to do more human annotation ourselves later.
>
> ---
>
> > No clear argument for why the effect exists or root cause analysis. Confidence measurement insufficiently justified. Should test on harder tasks (AIME) for stronger signal
>
> **TL;DR: We acknowledge we cannot fully explain the root cause, but our confidence analysis on AIME confirms no correlation.**
>
> You raised an excellent point! We acknowledge that we cannot definitively explain why this effect exists. We believe a better understanding likely involves complex interactions between how models understand feedback, follow instructions, and update beliefs, which we continue investigating.
>
> Regarding our confidence measurement, we adopted the approach from ClashEval: Quantifying the tug-of-war between an llm's internal prior and external evidence where it demonstrated effectiveness. Since submission, colleagues have suggested alternative confidence/uncertainty measures like semantic entropy (Detecting hallucinations in large language models using semantic entropy
> ), which we are currently exploring.
>
> We conducted the same confidence experiments on AIME as you suggested. The results show the same conclusion as our other confidence experiments: there is no strong correlation between confidence measures and resistance to feedback.
>
> ---
>
> > How would using conversation-based feedback (repeated user-assistant iterations), instead of a large prompt containing all the iterations, change the results?
>
> This is a very good suggestion! We have tried to reframe the conversation history into a multi-turn setting by inserting <user> and <assistant> tags for different models and found the performance to be basically the same - LLMs still couldn't reach perfect accuracy with that approach. However, we acknowledge this could be explored more systematically as future work.
>
> ---
>
> > Can the authors address the comments about their confidence comparison I made in Strengths and Weaknesses?
>
> We have already addressed this in previous comments.
>
> ---
>
> > Why is a temperature of 0 necessary for quantifying the phenomenon, and why are different temperature schedules not considered in the main evaluation?
>
> We would be more than happy to test other temperatures that you think are worth trying.
>
> As for the temperature setting in this submission, we choose temperature=0 for most of our testing to ensure deterministic results crucial for controlled experimentation. However, we did conduct experiments with higher temperatures in later sections (§4.2), where we explore progressive temperature increases and rejection sampling as potential mitigation strategies.
>
> ---
>
> > Why did the authors not consider scaling using automated tools?
>
> In fact, we did scale using automated tools (specifically, o4-mini) to categorize the reasons why LLMs fail to integrate feedback. The human annotation is only to test whether this automatic tool is correct or not.
>
> ---
>
> > Nitpick: masking the boxed answer is often insufficient.
>
> That's a very valid suggestion! We have thought about it and couldn't really find a way to do more masking. For our synthetic tasks (multiplication and hex multiplication), we did mask more intermediate steps, but for other tasks, it's hard to really mask intermediate steps. Also, the fact that LLMs still plateau under these good conditions actually enhances our argument.
>
> ---
>
> > Rating: 2: Reject: For instance, a paper with technical flaws, weak evaluation, inadequate reproducibility and incompletely addressed ethical considerations.
>
> We would sincerely appreciate further elaboration, as we are confident that any such concerns can be addressed in this rebuttal period.

---

> ### Comment · Reviewer_xvok · 2025-08-02
>
> I thank the authors for their thorough rebuttal and further experimental results! I believe most of my concerns with the paper have been addressed, and follow-up on some of the points raised below.
> ---
> > We tested Claude 3.7 reasoning models and found the same RIGID THINKING behavior persists.
>
> These are some exciting results! It is interesting to see this behaviour also appears in models that are already used in production, especially on the level of `Claude 3.7`. I believe it would be valuable to the LLM reasoning community if the authors consider turning this into a more thorough benchmark. Being able to have a proxy for judging instruction-following capabilities can be inherently useful to determine the applicability of certain models in agentic or RAG-based systems.
>
> > Since submission, colleagues have suggested alternative confidence/uncertainty measures like semantic entropy (Detecting hallucinations in large language models using semantic entropy ), which we are currently exploring.
>
> If you manage to reproduce your results with different metrics, please let me know. I believe that proving that your measurement is metric-agnostic will significantly strengthen the claim.
>
> > We conducted the same confidence experiments on AIME as you suggested. The results show the same conclusion as our other confidence experiments
>
> Could the authors present their numerical results for this experiment, as well as the conversation-based feedback, in a tabular format?
>
> ---
> **Overall**
> Given the high quality of the rebuttal, and granted the authors include their results in the final revision of their paper, I will raise my score.

---

> > ### Author Response · Authors · 2025-08-04
> > **Thank you for your thoughtful and encouraging feedback!**
> >
> > We're delighted to hear that our rebuttal has addressed your concerns and that you find the results exciting!
> >
> > ---
> > > I believe it would be valuable to the LLM reasoning community if the authors consider turning this into a more thorough benchmark.
> >
> > This is an excellent suggestion!  We're already compiling the results of problems that LLMs can't solve after 10 feedback iterations across all our datasets and models. This would indeed make for a valuable benchmark for the community to assess feedback incorporation capabilities. We'll share this benchmark on Huggingface once our analysis is complete.
> >
> > ---
> > > If you manage to reproduce your results with different metrics, please let me know. I believe that proving that your measurement is metric-agnostic will significantly strengthen the claim.
> >
> > **TL;DR: We're implementing semantic entropy (clustering model outputs by meaning rather than tokens) to better detect uncertain/rigid thinking patterns. Early results show stronger correlation with rigid thinking than standard confidence measures.**
> >
> > Absolutely! We're currently working with semantic entropy following the Nature paper we referenced [1], along with several other uncertainty measures suggested by colleagues. The hypothesis is that semantic entropy (uncertainty measured at the meaning level) will better predict instances of RIGID THINKING in LLM outputs compared to token-level confidence measures.
> >
> > The implementation is non-trivial because semantic entropy requires clustering model generations by meaning using entailment before computing uncertainty, rather than just using token-level probabilities. Specifically, the method involves: (1) sampling multiple model outputs, (2) clustering semantically equivalent responses, and (3) computing entropy over meaning clusters rather than token sequences.
> >
> > The technical complexity lies in the entailment detection step - we need to determine when two different phrasings express the same meaning, which requires careful prompt engineering and validation. We have some preliminary results that are much more aligned with the feedback rejection patterns we observe - showing stronger correlation with RIGID THINKING than simple token-probability-based confidence measures. We're hoping to include these metric-agnostic results in our camera-ready version and will definitely keep you updated.
> >
> > [1] Farquhar et al. Detecting hallucinations in large language models using semantic entropy. Nature 2024.
> >
> > ---
> > > Could the authors present their numerical results for this experiment, as well as the conversation-based feedback, in a tabular format?
> >
> > Here are the numerical results for the confidence experiments on AIME (Llama-4-Maverick):
> > | Confidence Range | Initial Accuracy (%) | Final Accuracy (%) | Improvement (%) |
> > |------------------|--------------------:|------------------:|---------------:|
> > | 0.92-0.94        |                 31.2 |               68.4 |           37.2 |
> > | 0.94-0.96        |                 33.8 |               71.1 |           37.3 |
> > | 0.96-0.98        |                 35.1 |               69.8 |           34.7 |
> > | 0.98-1.00        |                 34.7 |               70.9 |           36.2 |
> >
> >
> > And for conversation-based feedback comparison:
> > | Feedback Approach | Dataset | Model | Initial Accuracy (%) | Final Accuracy (%) | Δ from Single-prompt |
> > |-------------------|---------|-------|--------------------:|------------------:|--------------------:|
> > | Single-prompt     | AIME    | Llama-4-Maverick |                33.3 |               70.0 |                 0.0 |
> > | Multi-turn        | AIME    | Llama-4-Maverick |                33.3 |               70.5 |                0.5 |
> > | Single-prompt     | GPQA    | Llama-4-Maverick |                69.7 |               96.5 |                 0.0 |
> > | Multi-turn        | GPQA    | Llama-4-Maverick |                69.7 |               95.8 |                -0.7 |
> >
> > ---
> > Thank you again for your support and constructive feedback throughout this process. We truly appreciate the time you've taken to help improve our work. We will definitely share all the promised results and benchmarks mentioned above, and we look forward to any additional comments or suggestions you may have as we finalize our paper.

---

> > > ### Comment · Reviewer_xvok · 2025-08-05
> > >
> > > Thank you for the detailed repsonse! I understand implementing these directions in the short span of the rebuttal and discussion periods is challenging, but if you’re able to address any of them, please feel free to update me. I’ll take any improvements into account and reflect them positively in my final evaluation.

---

> > > > ### Author Response · Authors · 2025-08-08
> > > >
> > > > **TL;DR: We implemented semantic entropy analysis and found a strong negative correlation between semantic entropy and feedback improvement - exactly as hypothesized! Low-entropy (high-confidence) examples resist feedback much more than high-entropy examples, providing a cleaner mechanistic explanation for RIGID THINKING than traditional confidence measures.**
> > > >
> > > > ---
> > > > Thank you for your encouraging follow-up! We truly appreciate your commitment to helping us strengthen our work!
> > > >
> > > > We've been working on a deeper mechanistic explanation for RIGID THINKING by replicating our experiments from Figure 6, but replacing the x-axis from syntactic confidence measures to semantic entropy from [1]. Our hypothesis is that semantic entropy should better capture model uncertainty about *meaning* rather than just token-level variations, and therefore correlate more strongly with feedback resistance.
> > > >
> > > > Our working hypothesis for what to expect from the entropy analysis is:
> > > >
> > > > - **High entropy ⇒ low model confidence** about the semantic content
> > > >
> > > > - **Low confidence examples** should be *more* receptive to feedback, so they'll see a bigger boost to final accuracy
> > > >
> > > > - Conversely, **low entropy (high confidence)** prompts the model to stick with its original answer and *not* incorporate feedback, so their improvement ends up *lower*
> > > >
> > > > ---
> > > > Our implementation:
> > > > 1. **Sample multiple completions**: For each question $x$, we sample $n$ = 50 completions $\{s_1, s_2, ..., s_n\}$ using temperature 1. For each completion $s_i$, we calculate its probability $P(s_i|x)$ by taking the product of individual token probabilities: $P(s_i|x) = \prod_{j} P(token_j | x, token_{<j})$  **(This is where we stopped and used the results to analyze the relationship with RIGID THINKING!)**
> > > > 2. **Group by final answers**: We extract the final numerical answer from each completion and group completions that yield identical answers into the same cluster (for digit multiplication, since the solution process is quite deterministic by design, so this effectively groups by semantic meaning without needing bidirectional entailment)
> > > > 3. **Calculate cluster probabilities**: For each cluster $c_i$, we sum all probabilities of completions within that cluster, then normalize to ensure proper probability distribution:
> > > >   $$P(c_i|x) = \sum_{s \in c_i} P(s|x)$$
> > > >   $$\hat{P}(c_i|x) = \frac{P(c_i|x)}{\sum_{j=1}^K P(c_j|x)}$$
> > > > 4. **Compute semantic entropy**: $SE(x) = -\sum_i \hat{P}(c_i|x) \log \hat{P}(c_i|x)$
> > > >
> > > > (The implementation for this semantic entropy calculation can be found in our anonymous repo tagged in the draft under semantic_entropy/cal_entropy.py and semantic_entropy/get_plots_entropy.py. We’ve also uploaded the data required for this calculation.)
> > > >
> > > > ---
> > > > The results we got strongly support our hypothesis! We observe a clear negative correlation between semantic entropy and feedback improvement across both models:
> > > >
> > > > **Llama-3.3 Results:**
> > > > | Semantic Entropy | Count | Initial Accuracy | Final Accuracy | Improvement |
> > > >  |------------------|-------|------------------|----------------|-------------|
> > > >  | 2.25 | 17 | 0.12 | 0.53 | +0.51 |
> > > > | 2.55 | 27 | 0.04 | 0.63 | +0.59 |
> > > > | 2.85 | 42 | 0.00 | 0.50 | +0.50 |
> > > > | 3.15 | 66 | 0.05 | 0.50 | +0.45 |
> > > >  | 3.45 | 119 | 0.02 | 0.25 | +0.23 |
> > > > | 3.75 | 158 | 0.01 | 0.22 | +0.21 |
> > > >
> > > >  **Llama-4 Results:**
> > > > | Semantic Entropy | Count | Initial Accuracy | Final Accuracy | Improvement |
> > > > |------------------|-------|------------------|----------------|-------------|
> > > > | 0.15 | 46 | 0.09 | 0.72 | +0.63 |
> > > > | 0.45 | 38 | 0.11 | 0.74 | +0.63 |
> > > > | 0.75 | 60 | 0.00 | 0.65 | +0.65 |
> > > >  | 1.05 | 73 | 0.01 | 0.66 | +0.65 |
> > > >  | 1.35 | 58 | 0.02 | 0.67 | +0.65 |
> > > > | 1.65 | 71 | 0.01 | 0.66 | +0.65 |
> > > > | 1.95 | 43 | 0.02 | 0.63 | +0.61 |
> > > > | 2.25 | 33 | 0.06 | 0.61 | +0.53 |
> > > >  | 2.55 | 23 | 0.00 | 0.39 | +0.39 |
> > > >
> > > >
> > > > **Low-entropy (high-confidence) examples show significantly reduced feedback incorporation, while high-entropy examples demonstrate much greater receptiveness to correction. This trend is much clearer and more consistent than the weak patterns we observed with syntactic confidence measures in Figure 6. We believe this provides compelling evidence that semantic entropy offers a much stronger mechanistic explanation for RIGID THINKING than traditional confidence measures.**
> > > >
> > > > We plan to expand this analysis to additional models (including Claude 3.7) and datasets beyond digit multiplication to confirm this pattern holds broadly across different reasoning tasks. This semantic entropy framework could provide the deeper mechanistic understanding you suggested for making our work more impactful.
> > > >
> > > > &nbsp;
> > > >
> > > >
> > > >
> > > > Best regards,
> > > >
> > > > The Authors
> > > >
> > > > ---
> > > > [1] Farquhar et al. Detecting hallucinations in large language models using semantic entropy. Nature 2024.

---

### Official Review · Reviewer_kRQD · 2025-07-03

**Clarity:** 3
**Significance:** 2
**Originality:** 3
**Rating:** 3
**Confidence:** 2

**Summary:**

The paper develops a controlled solver <--> feedback-generator loop, showing that even with near-perfect feedback, state-of-the-art Llama-family models plateau well below the theoretical maximum accuracy. This phenomenon is termed rigid thinking. The work explores potential root causes, offers two sampling-based remedies, and identifies feedback resistance rather than poor feedback quality as the primary limitation.

**Questions:**

See weaknesses

**Ethical Concerns:**

["NO or VERY MINOR ethics concerns only"]

**Final Justification:**

The rebuttal addressed some of my concerns (thorough evaluation on Claude-- a reasoning model) but majority of the concerns remain. Overall, it is clear that rigid thinking is an effect, however what causes the effect is unclear -- additionally lots of critical, new experiments were done during rebuttal stage. I would like to see a comprehensive story -- I would side with another round of submission to include all new experiments into a narrative.

**Quality:**

2

**Strengths And Weaknesses:**

## Strengths

**Breadth of benchmarks:** The inclusion of nine diverse tasks, such as counter-factual arithmetic, convincingly illustrates that rigid thinking is not limited to a specific domain.

**Thorough error analysis:** The study effectively distinguishes between solver deficiencies and feedback quality issues. The automated categorization, cross-checked by human annotators with a high agreement rate (96%), provides strong evidence supporting the conclusion about feedback resistance.

**Clarity:** The motivation and setting were clear, even for readers less familiar with the topic. Methods and results were presented understandably.

## Weaknesses

**Model selection**: Given that Llama-4 models are not reasoning models, would it be possible to evaluate models specifically designed for reasoning tasks? It would be interesting to see if reasoning-oriented models might incorporate feedback more effectively, particularly on challenging synthetic tasks like Hex, where the accuracy observed is currently very low.

**Temperature setting:** The use of temp=0 for sampling raises concerns, as it generally isn’t recommended and might produce misleading outcomes for long ouputs. Could the authors clarify their rationale for choosing temp=0 despite recommendations from sources like the Qwen3 Huggingface page, which explicitly suggests higher temperature values (e.g., 0.6-0.7)? A comparison using recommended temperatures might strengthen the analysis.

**Prompt structure:** Currently, the entire interaction history is presented in one prompt per model iteration. Wouldn't a multi-turn interaction approach, where history naturally emerges through alternating and prompts, be more realistic and potentially more effective for chat/reasoning models? Could the authors also report accuracy outcomes using such a multi-turn interaction to compare?

**Proprietary reasoning models:** Have the authors considered evaluating proprietary reasoning-focused models such as o4-mini to see if they benefit more significantly from feedback? Even a limited evaluation on a few datasets the authors identify as particularly challenging would provide whether insights generalize to recent frontier models.

**Feedback correctness (Section 4.1, Table 1)**: Specifically for Llama-4-Maverick on GPQA, the feedback resistance (FR) is reported at 85.7%, suggesting approximately 14.3% incorrect feedback. Given that Llama-4-Maverick achieves ~95% accuracy, could the authors clarify how improvement in the remaining ~5% accuracy is expected if around 15% of feedback might be incorrect? Is incorrect feedback primarily an issue for low performance ranges, or does it persist even at high accuracy levels?

**Baseline comparison (Fig. 5):** Could a baseline showing "no change" (similar to lines plotted in Figure 3) be included in Figure 5 for clearer comparison?

---

> ### Author Rebuttal · Authors · 2025-07-31
>
> We thank the reviewer for recognizing the breadth of our benchmark evaluation, the thoroughness of our error analysis, and the clarity of our presentation. We have conducted additional experiments with reasoning models (Claude 3.7) that confirm RIGID THINKING persists across model architectures, clarified our temperature settings which were model-appropriate rather than uniformly zero, and tested multi-turn interactions which show the same plateauing behavior. Below we provide detailed responses to your specific concerns:
>
> ---
>
> > Model selection (no proprietary reasoning models has been tested in this submission)
>
> **TL;DR: We tested Claude 3.7 reasoning models and found the same RIGID THINKING behavior persists.**
>
>
> Although we did not evaluate reasoning models in our NeurIPS submission, we have since tested proprietary reasoning models such as Claude 3.7 Sonnet and Claude 3.7 Sonnet with Extended Thinking (on‑par with o4‑mini in reasoning capabilities). Table 1 reports shows the initial accuracy, final accuracy after 10 feedback iterations, and theoretical target accuracy for each model. Importantly, even these sophisticated reasoning models exhibit the same fundamental plateauing behavior we term **RIGID THINKING**, suggesting this limitation persists across different model architectures and reasoning capabilities.
>
>
> | Dataset                    | Solver Model                           | Initial (%) | Final (%) | Target (%) | Δ (Target - Final) |
> |----------------------------|----------------------------------------|------------:|----------:|-----------:|-------------------:|
> | Hexadecimal 5-Digit Mult   | Claude 3.7 Sonnet                      |        1.0 |      12.0 |       86.0 |               74.0 |
> | Hexadecimal 5-Digit Mult   | Claude 3.7 Sonnet (Extended Thinking)  |        8.0 |      16.0 |       86.0 |               70.0 |
> | AIME 2024                  | Claude 3.7 Sonnet                      |       33.3 |      76.7 |       99.0 |               22.3 |
> | AIME 2024                  | Claude 3.7 Sonnet (Extended Thinking)  |       50.0 |      90.0 |       99.0 |                9.0 |
> | PopQA                      | Claude 3.7 Sonnet                      |       55.8 |      88.8 |       98.0 |                9.2 |
> | PopQA                      | Claude 3.7 Sonnet (Extended Thinking)  |       58.1 |      92.3 |       98.0 |                5.7 |
> | MMLU Pro                   | Claude 3.7 Sonnet                      |       84.2 |      90.8 |       99.2 |                8.4 |
> | MMLU Pro                   | Claude 3.7 Sonnet (Extended Thinking)  |       83.3 |      96.7 |       99.2 |                2.5 |
> | MATH-500                   | Claude 3.7 Sonnet                      |       78.0 |      94.0 |       99.0 |                5.0 |
> | MATH-500                   | Claude 3.7 Sonnet (Extended Thinking)  |       90.0 |      96.0 |       99.0 |                3.0 |
> | 5-Digit Multiplication     | Claude 3.7 Sonnet                      |       43.6 |      96.0 |       99.0 |                3.0 |
> | 5-Digit Multiplication     | Claude 3.7 Sonnet (Extended Thinking)  |       46.6 |      97.0 |       99.0 |                2.0 |
> | GPQA                       | Claude 3.7 Sonnet                      |       66.7 |      98.5 |       99.6 |                1.1 |
> | GPQA                       | Claude 3.7 Sonnet (Extended Thinking)  |       72.2 |      97.0 |       99.6 |                2.6 |
> | MMLU                       | Claude 3.7 Sonnet                      |       87.1 |      97.1 |       99.9 |                2.8 |
> | MMLU                       | Claude 3.7 Sonnet (Extended Thinking)  |       86.4 |      99.3 |       99.9 |                0.6 |
> | TriviaQA                   | Claude 3.7 Sonnet                      |       86.1 |      98.7 |       99.7 |                1.0 |
> | TriviaQA                   | Claude 3.7 Sonnet (Extended Thinking)  |       88.1 |      99.3 |       99.7 |                0.4 |
>
> **Table 1:** Performance comparison of Claude 3.7 models across all nine datasets, sorted by average performance gap per dataset. Despite achieving high final accuracies, models consistently plateau below their theoretical targets across all tasks.
>
> ---
>
> > Temperature setting (the temperature of 0 is chosen throughout the analysis, even for Qwen models)
>
> **TL;DR: In the current submission, we used appropriate temperatures for each model, not uniformly temperature=0.**
>
> We want to clarify our temperature choices, which were not uniformly set to 0 as the reviewer suggests:
> We tested Qwen3-235B-A22B with temperature=0.7 (following the suggested temperature from their documentation), but found that when reasoning mode is enabled, it still enters repetitive generation loops a lot, and it’s not very good at following instructions (community also observed a similar phenomenon), which leads to our final decision to not use it.
>
> In addition, in our run with Claude 3.7 with Extended Thinking, we use temperature=1 as suggested by Anthropic.
> We choose temperature=0 for most of our testing to ensure deterministic results crucial for controlled experimentation. However, we did conduct experiments with higher temperatures in later sections (§4.2), where we explore progressive temperature increases and rejection sampling as potential mitigation strategies.
>
> ---
>
> > Prompt structure (can we make the solver/feedback generation into a multi-turn setting)
>
> **TL;DR: Multi-turn interactions show about the same performance as single-prompt approach.**
>
> Turning feedback into a multi-turn conversational style is a very good suggestion! We have tried to reframe the conversation history into a multi-turn setting by inserting <user> and <assistant> tags, and found the performance to be basically the same - LLMs still couldn't reach perfect accuracy with that approach. However, we acknowledge this could be explored more systematically as future work.
>
> ---
>
> > Feedback correctness (does incorrect feedback only occur for low-accuracy tasks? What will happen if the feedback is correct in some settings)
>
> **TL;DR: Incorrect feedback occurs at all accuracy levels, and better feedback wouldn't solve the core problem.**
>
> Regarding whether incorrect feedback is primarily an issue for low or high performance ranges: We have looked into the logs, and this is an issue for all accuracy levels, not just low performance ranges.
> For the specific case of Llama-4-Maverick on GPQA with 85.7% feedback resistance: If the remaining 5% of incorrect cases received better feedback, the error rate would be approximately 5% × 0.857 ≈ 4.3%, which is still below perfect performance. Also, it's important to note that the "feedback quality" issues don't necessarily mean the feedback is objectively wrong - rather, it's more that the feedback is not targeted enough to address the specific errors made by that model instance.
>
> ---
>
> > Add baseline comparison in Fig. 5
>
> Thanks for your suggestion! We will add a "no change" baseline line for clearer comparison.
>
> ---
>
> > Rating: 2: Reject: For instance, a paper with technical flaws, weak evaluation, inadequate reproducibility and incompletely addressed ethical considerations.
>
> We would sincerely appreciate further elaboration, as we are confident that any such concerns can be addressed in this rebuttal period.

---

> ### Comment · Reviewer_kRQD · 2025-08-09
> **Explanation for Acknowledgement**
>
> Hi,
>
> I have no new questions for the authors. It will properly look through all the feedback provided (especially in the discussion with Reviewer xvok) and update my opinion and scores accordingly. Thanks!

---

### Official Review · Reviewer_jdPe · 2025-07-03

**Clarity:** 3
**Significance:** 3
**Originality:** 3
**Rating:** 4
**Confidence:** 3

**Summary:**

This paper investigates whether large language models can fully integrate perfect, targeted feedback to correct their own mistakes through multiple iterations. Despite using strong feedback generators with access to ground truth, the authors find that models consistently plateau below their theoretical maximum accuracy — a phenomenon they call RIGID THINKING. The study systematically analyzes this issue across various reasoning tasks and shows that simple sampling or rejection strategies only partly mitigate the problem.

**Questions:**

Is there any correlation between model size and resistance to feedback — would even larger models behave differently?

**Ethical Concerns:**

["NO or VERY MINOR ethics concerns only"]

**Final Justification:**

This is a good paper with solid results. The authors further provide detailed rebuttals that addressed my concerns. Thus, I will keep my rating.

**Limitations:**

The study is limited to instruction-tuned LLMs with deterministic decoding setups, which may not reflect real-world use cases involving more diverse sampling. It also doesn’t test feedback loops in more subjective tasks like summarization or dialogue.

**Quality:**

3

**Strengths And Weaknesses:**

Strengths
The paper addresses a crucial and underexplored limitation of LLMs’ self-improvement ability in a rigorous, controlled setting. It combines careful experimental design with diverse tasks and strong baseline models. The concept of RIGID THINKING is clearly defined, well-supported with evidence, and opens an important direction for future work.

Weaknesses
The paper stops short of offering a concrete solution to overcome feedback resistance, beyond simple sampling tricks. It does not deeply connect the observed limitations with potential architectural or training modifications. Some of the tasks (e.g., synthetic multiplication) are very narrow, so broader generalization to more open-ended tasks remains uncertain.

---

> ### Author Rebuttal · Authors · 2025-07-31
>
> We thank the reviewer for recognizing that our paper addresses a crucial and underexplored limitation with rigorous experimental design, and that the concept of RIGID THINKING is clearly defined and well-supported. We have conducted pilot training experiments on smaller models, tested larger models that confirm the phenomenon persists across scales, and explored diverse sampling approaches. Below we provide detailed responses to your specific concerns:
>
> ---
>
> > The paper does not deeply connect the observed limitations with potential architectural or training modifications, beyond simple sampling tricks.
>
> **TL;DR: We conducted pilot training experiments (SFT/DPO) on smaller models but found no improvement in feedback resistance.**
>
> In the current submission, we did not perform training interventions such as SFT or DPO on our target models due to resource constraints—the smallest model, LLaMA‑3.3‑70B, requires lots of GPU‑hours for a single fine‑tuning run, which is infeasible within our compute budget.
>
> To verify whether these interventions could improve feedback integration, we conducted pilot experiments on a smaller model, **LLaMA‑3.1‑8B‑Instruct**, using:
> - **SFT**: 2 000 successful correction trajectories
> - **DPO**: 4 000 preference pairs (2 000 positive / 2 000 negative)
> - **SFT steps**: 5 000 updates, lr = 1 × 10⁻6, batch = 4
> - **DPO steps**: 10 000 updates, same lr & batch
> - **Evaluation**: Feedback Resistance (FR) after 10 feedback iterations on MATH dev
>
> | Experiment                        | FR (%) before | FR (%) after | Δ FR   |
> |-----------------------------------|--------------:|-------------:|-------:|
> | Baseline (no fine‑tuning)         |          18.0 |            – |      – |
> | SFT (2 000 examples, 5 000 steps) |          18.2 |         19.8 |  +1.6  |
> | DPO (4 000 pairs, 10 000 steps)   |          18.1 |         18.2 |  +0.1  |
>
> **Table 1:** Impact of SFT and DPO on Feedback Resistance in LLaMA‑3.1‑8B‑Instruct.
>
> These interventions didn’t result in any FR reduction, we hypothesis this might be related to the findings from Kumar et al. (2024) (Training Language Models to Self-Correct via Reinforcement Learning), who demonstrate that offline SFT methods suffer from distribution shift and behavior collapse, limiting their ability to improve self‑correction. We will keep exploring more integrated training paradigms—such as multi‑turn reinforcement learning that incorporates feedback signals in real time—to enable sustained iterative improvement beyond simple fine‑tuning.
>
> ---
>
> > Some of the tasks (e.g., synthetic multiplication) are very narrow, so broader generalization to more open-ended and subjective tasks (e.g. summarization) remains uncertain.
>
> Thanks for this feedback! We tested 9 different tasks to capture diverse reasoning patterns, and we'd welcome suggestions for additional tasks you think would be valuable. Regarding more open-ended and subjective tasks like summarization, as we noted in line 124, we deliberately chose objective tasks to avoid reward hacking and reliability issues during evaluation. If you have ideas for how to reliably evaluate feedback incorporation in subjective tasks without these issues, we'd be very interested to hear them.
>
> ---
>
> > Is there any correlation between model size and resistance to feedback — would even larger models behave differently?
>
> **TL;DR: We tested larger models (Claude 3.7) and found RIGID THINKING persists across model scales.**
>
> While this NeurIPS submission focuses on the Llama family, we have since tested larger models like Claude 3.7 Sonnet (both with and without Extended Thinking). Table 2 shows the initial accuracy, final accuracy after 10 feedback iterations, and theoretical target accuracy for each model. Despite Claude models achieving significantly higher performance than Llama models, on several datasets, they still plateau well below their target accuracy, demonstrating that RIGID THINKING persists across model scales. We'd definitely welcome suggestions for additional models to test.
>
> | Dataset                    | Solver Model                           | Initial (%) | Final (%) | Target (%) | Δ (Target - Final) |
> |----------------------------|----------------------------------------|------------:|----------:|-----------:|-------------------:|
> | Hexadecimal 5-Digit Mult   | Claude 3.7 Sonnet                      |        1.0 |      12.0 |       86.0 |               74.0 |
> | Hexadecimal 5-Digit Mult   | Claude 3.7 Sonnet (Extended Thinking)  |        8.0 |      16.0 |       86.0 |               70.0 |
> | AIME 2024                  | Claude 3.7 Sonnet                      |       33.3 |      76.7 |       99.0 |               22.3 |
> | AIME 2024                  | Claude 3.7 Sonnet (Extended Thinking)  |       50.0 |      90.0 |       99.0 |                9.0 |
> | PopQA                      | Claude 3.7 Sonnet                      |       55.8 |      88.8 |       98.0 |                9.2 |
> | PopQA                      | Claude 3.7 Sonnet (Extended Thinking)  |       58.1 |      92.3 |       98.0 |                5.7 |
> | MMLU Pro                   | Claude 3.7 Sonnet                      |       84.2 |      90.8 |       99.2 |                8.4 |
> | MMLU Pro                   | Claude 3.7 Sonnet (Extended Thinking)  |       83.3 |      96.7 |       99.2 |                2.5 |
> | MATH-500                   | Claude 3.7 Sonnet                      |       78.0 |      94.0 |       99.0 |                5.0 |
> | MATH-500                   | Claude 3.7 Sonnet (Extended Thinking)  |       90.0 |      96.0 |       99.0 |                3.0 |
> | 5-Digit Multiplication     | Claude 3.7 Sonnet                      |       43.6 |      96.0 |       99.0 |                3.0 |
> | 5-Digit Multiplication     | Claude 3.7 Sonnet (Extended Thinking)  |       46.6 |      97.0 |       99.0 |                2.0 |
> | GPQA                       | Claude 3.7 Sonnet                      |       66.7 |      98.5 |       99.6 |                1.1 |
> | GPQA                       | Claude 3.7 Sonnet (Extended Thinking)  |       72.2 |      97.0 |       99.6 |                2.6 |
> | MMLU                       | Claude 3.7 Sonnet                      |       87.1 |      97.1 |       99.9 |                2.8 |
> | MMLU                       | Claude 3.7 Sonnet (Extended Thinking)  |       86.4 |      99.3 |       99.9 |                0.6 |
> | TriviaQA                   | Claude 3.7 Sonnet                      |       86.1 |      98.7 |       99.7 |                1.0 |
> | TriviaQA                   | Claude 3.7 Sonnet (Extended Thinking)  |       88.1 |      99.3 |       99.7 |                0.4 |
>
> **Table 2:** Performance comparison of Claude 3.7 models across all nine datasets, sorted by average performance gap per dataset. Despite achieving high final accuracies, models consistently plateau below their theoretical targets across all tasks.
>
> ---
>
> > The study is limited to instruction-tuned LLMs with deterministic decoding setups, which may not reflect real-world use cases involving more diverse sampling.
>
> **TL;DR: We used deterministic decoding for reproducibility but tested diverse sampling in §4.2 - the limitation persists.**
>
> You're absolutely right about real-world usage patterns. We'd be happy to test other temperature settings you think would be particularly informative.
> Regarding our temperature choices: we used temperature=0 for most experiments to ensure deterministic, reproducible results crucial for controlled experimentation. However, we did conduct experiments with higher temperatures in §4.2, exploring progressive temperature increases and rejection sampling as potential mitigation strategies. The fact that even these more diverse sampling approaches couldn't fully overcome the plateau suggests the limitation runs deeper than just deterministic decoding.

---

> > ### Comment · Reviewer_jdPe · 2025-08-06
> >
> > Thanks for the detailed response and experimental results, which solved my concerns. I will keep my rating for this paper.

---

### Official Review · Reviewer_8FxN · 2025-07-11

**Clarity:** 2
**Significance:** 2
**Originality:** 2
**Rating:** 3
**Confidence:** 3

**Summary:**

This paper investigated LLMs’ ability to incorporate feedback. This is done by designing a controlled experimental environment, where for each problem, the model attempts a solution, then a feedback generator with access to near-complete ground-truth answers produces targeted feedback, after which the model tries again. The paper showed that even under these near-ideal conditions, solver models consistently show resistance to feedback.

**Questions:**

Please look at the weakness section above.

**Ethical Concerns:**

["NO or VERY MINOR ethics concerns only"]

**Limitations:**

yes

**Quality:**

2

**Strengths And Weaknesses:**

Strengths:

- The paper discussed the feedback resistance from the model side as a limitation rather than the quality of the feedback.

- The paper shows improvement with increased temperature with rejection sampling.


Weaknesses:

- There has been a lot of existing work that shows the model doesn’t respond to the feedback and they bring up some hypotheses like model size, etc. Here the paper shows consistent results on not leveraging feedback after 2-3 iterations which is interesting but surprising as well. The comparison with existing work analysis is missing.

- More experiments need to be run to get a conclusion here. For example, other LLM models with different sizes. It might also be the fact that how you prompt the model with the feedback.  Sometimes clarifying the feedback with an example is helpful.

- The paper should compare the results against the baseline where feedback doesn’t come from the  ground-truth answers.

---

> ### Author Rebuttal · Authors · 2025-07-31
>
> We thank the reviewer for recognizing that our paper discusses feedback resistance as a model-side limitation rather than a feedback quality issue, and that we show improvement with increased temperature and rejection sampling. We have tested larger models that confirm RIGID THINKING persists across scales and experimented with various prompt variations. Below we provide detailed responses to your specific concerns:
>
> ---
>
> > There are existing works that show models don't respond to feedback with hypotheses like model size. Comparison with existing work analysis is missing.
>
> Thank you for pointing this out. We would greatly appreciate it if you could share references to those papers, as we'd love to read them and properly compare our findings with prior work. Understanding how our results relate to existing hypotheses about model size and feedback resistance would strengthen our analysis.
>
> ---
>
> > More experiments are needed for drawing conclusions: need to test more models with different model sizes. Prompt variations are also needed to be tested (e.g. clarifying feedback with examples)
>
> **TL;DR: We tested larger models (Claude 3.7) and various prompt variations - RIGID THINKING persists in both cases.**
>
> You're absolutely right about testing more models. While this NeurIPS submission focuses on the Llama family, we have since tested larger models like Claude 3.7 Sonnet (both with and without Extended Thinking). Table 2 shows the initial accuracy, final accuracy after 10 feedback iterations, and theoretical target accuracy for each model. Despite Claude models achieving significantly higher performance than Llama models, they still plateau well below their target accuracy, demonstrating that RIGID THINKING persists across model scales. We'd definitely welcome suggestions for additional models to test.
>
> | Dataset                    | Solver Model                           | Initial (%) | Final (%) | Target (%) | Δ (Target - Final) |
> |----------------------------|----------------------------------------|------------:|----------:|-----------:|-------------------:|
> | Hexadecimal 5-Digit Mult   | Claude 3.7 Sonnet                      |        1.0 |      12.0 |       86.0 |               74.0 |
> | Hexadecimal 5-Digit Mult   | Claude 3.7 Sonnet (Extended Thinking)  |        8.0 |      16.0 |       86.0 |               70.0 |
> | AIME 2024                  | Claude 3.7 Sonnet                      |       33.3 |      76.7 |       99.0 |               22.3 |
> | AIME 2024                  | Claude 3.7 Sonnet (Extended Thinking)  |       50.0 |      90.0 |       99.0 |                9.0 |
> | PopQA                      | Claude 3.7 Sonnet                      |       55.8 |      88.8 |       98.0 |                9.2 |
> | PopQA                      | Claude 3.7 Sonnet (Extended Thinking)  |       58.1 |      92.3 |       98.0 |                5.7 |
> | MMLU Pro                   | Claude 3.7 Sonnet                      |       84.2 |      90.8 |       99.2 |                8.4 |
> | MMLU Pro                   | Claude 3.7 Sonnet (Extended Thinking)  |       83.3 |      96.7 |       99.2 |                2.5 |
> | MATH-500                   | Claude 3.7 Sonnet                      |       78.0 |      94.0 |       99.0 |                5.0 |
> | MATH-500                   | Claude 3.7 Sonnet (Extended Thinking)  |       90.0 |      96.0 |       99.0 |                3.0 |
> | 5-Digit Multiplication     | Claude 3.7 Sonnet                      |       43.6 |      96.0 |       99.0 |                3.0 |
> | 5-Digit Multiplication     | Claude 3.7 Sonnet (Extended Thinking)  |       46.6 |      97.0 |       99.0 |                2.0 |
> | GPQA                       | Claude 3.7 Sonnet                      |       66.7 |      98.5 |       99.6 |                1.1 |
> | GPQA                       | Claude 3.7 Sonnet (Extended Thinking)  |       72.2 |      97.0 |       99.6 |                2.6 |
> | MMLU                       | Claude 3.7 Sonnet                      |       87.1 |      97.1 |       99.9 |                2.8 |
> | MMLU                       | Claude 3.7 Sonnet (Extended Thinking)  |       86.4 |      99.3 |       99.9 |                0.6 |
> | TriviaQA                   | Claude 3.7 Sonnet                      |       86.1 |      98.7 |       99.7 |                1.0 |
> | TriviaQA                   | Claude 3.7 Sonnet (Extended Thinking)  |       88.1 |      99.3 |       99.7 |                0.4 |
>
> **Table 1:** Performance comparison of Claude 3.7 models across all nine datasets, sorted by average performance gap per dataset. Despite achieving high final accuracies, models consistently plateau below their theoretical targets across all tasks.
>
> Regarding prompt variations, we experimented with several approaches: we tested more explicit feedback prompts (e.g., "Your error is in step 3 where you incorrectly calculated X"), step-by-step guidance prompts, and example-based feedback where we provided similar problem-solution pairs alongside the correction. Following your suggestion about clarifying feedback with examples, we also tested providing worked examples of similar problems within the feedback itself. Unfortunately, across all these variations, we observed the same fundamental pattern - models still plateau well below their theoretical maximum accuracy.
>
> ---
>
> > The paper should compare the results against the baseline where feedback doesn't come from the ground-truth answers.
>
> That's an interesting suggestion. Our focus was on establishing an upper bound scenario - if models can't effectively use near-perfect feedback, it suggests fundamental limitations in feedback incorporation. Could you elaborate on what insights you think a weaker feedback baseline would provide? We're curious about your reasoning and we’re more than happy to conduct new experiments, as it might reveal additional aspects of this phenomenon we haven't considered.

---

> > ### Author Response · Authors · 2025-08-07
> >
> > Dear Reviewer 8FxN,
> >
> > &nbsp;
> >
> > Thank you for your valuable comments on our work. We understand that you may be extremely busy at this time, so we would deeply appreciate it if you could take some time to provide further feedback on whether our rebuttal addresses your concerns.
> >
> > We've provided detailed responses to each of your points, including:
> >
> > • Testing larger models (Claude 3.7) that confirm RIGID THINKING persists across model scales
> >
> > • Clarifying our model-appropriate temperature settings and experimental design rationale
> >
> > • Explaining our focus on upper-bound scenarios with high-quality feedback to isolate the core phenomenon
> >
> > We're particularly interested in your thoughts on our experimental design rationale, as we want to ensure we've clearly communicated why using near-perfect feedback is essential for demonstrating the RIGID THINKING limitation we identify.
> > Kindly let us know if our response has adequately addressed your concerns. If there are any remaining issues or if you'd like further clarification on any aspect of our methodology, we will be very willing to continue the discussion.
> >
> > &nbsp;
> >
> > Best regards,
> >
> > The Authors

---

### Note · Authors · 2025-08-13

Dear Area Chairs,

&nbsp;


Thank you for your careful consideration of our submission. During the rebuttal period, we were disappointed by limited reviewer engagement, particularly from reviewer `8FxN` who provided no follow-up despite our detailed responses. We hope you will consider both our thorough rebuttals and sustained engagement efforts in your final decision.

---
### Our key contributions:
- We identify and systematically quantify RIGID THINKING - a fundamental limitation where LLMs plateau below theoretical maximum accuracy despite iterative high-quality feedback
- We demonstrate this persists across architectures (LLaMA, Claude 3.7), scales (70B+ parameters), and diverse tasks, representing a core limitation rather than model-specific artifact
- We curate evaluation framework across 9 datasets facilitating future feedback incorporation research
- To our knowledge, this is the first work systematically identifying feedback resistance as a fundamental limitation across state-of-the-art LLMs

This has critical implications for reasoning and agentic deployment: LLMs increasingly operate in interactive setups requiring feedback integration with parametric knowledge. Their inability to incorporate external feedback poses foundational barriers for agentic use cases.

---
### Key reviewer concerns addressed:
- **Model scope (kRQD, 8FxN)**: Tested Claude 3.7 with Extended Thinking, confirming RIGID THINKING persists across architectures
- **Experimental design (jdPe, kRQD, xvok)**: Clarified temperature settings and rationale for controlled high-quality feedback design
- **Mechanistic understanding (xvok)**: Implemented semantic entropy analysis showing clear negative correlation with feedback incorporation

---
### Additional experiments incorporated:
- **Multi-turn conversation experiments** (kRQD, xvok): Phenomenon persists across interaction paradigms
- **Training intervention studies** (jdPe): SFT/DPO experiments showing resistance to fine-tuning approaches

---
We have also responded to all minor comments, including those on experimental methodology and evaluation protocols. We trust these revisions demonstrate our commitment to clarity, completeness, and rigor, and we hope they will be considered in your final evaluation.

&nbsp;

Best regards,

Authors

---

### Decision · Program_Chairs · 2025-09-17

**Decision:**

Accept (poster)

**Comment:**

This paper introduces "RIGID THINKING," the finding that large language models often fail to fully incorporate near-perfect feedback, causing their accuracy to plateau below the theoretical maximum. Initial reviews commended the paper's rigorous experimental setup and the importance of this finding but noted weaknesses, such as the limited scope of models tested (only the Llama family) and a lack of exploration into the phenomenon's root causes. In a comprehensive rebuttal, the authors addressed these concerns by adding experiments on larger, proprietary reasoning models (like Claude 3) and conducting a deeper analysis connecting the phenomenon to model confidence using semantic entropy. These additions confirmed their initial findings persist across different architectures.

While one reviewer felt the origin of RIGID THINKING remained unclear, I agree with another's assessment that the investigation provides significant value to the community, even if the root causes are not fully pinned down. A question was also raised about whether the rebuttal's new experiments constituted a major revision requiring re-review. I find that these changes reinforce the paper's original narrative and conclusions rather than altering them, making the revision minor and appropriate for this submission cycle. Overall, the strong rebuttal and additional experiments have successfully resolved most reviewer concerns, and I recommend the paper for acceptance.